# Fine-mapping *cis*-regulatory variants in diverse human populations

**Ashley Tehranchi[1], Brian Hie[2], Michael Dacre[1], Irene Kaplow[2], Kade Pettie[1], Peter Combs[1], Hunter B Fraser[1]\***

[1]Department of Biology, Stanford University, Stanford, United States; [2]Department of Computer Science, Stanford University, Stanford, United States

**Abstract** Genome-wide association studies (GWAS) are a powerful approach for connecting genotype to phenotype. Most GWAS hits are located in cis-regulatory regions, but the underlying causal variants and their molecular mechanisms remain unknown. To better understand human *cis*-regulatory variation, we mapped quantitative trait loci for chromatin accessibility (caQTLs)—a key step in cis-regulation—in 1000 individuals from 10 diverse populations. Most caQTLs were shared across populations, allowing us to leverage the genetic diversity to fine-map candidate causal regulatory variants, several thousand of which have been previously implicated in GWAS. In addition, many caQTLs that affect the expression of distal genes also alter the landscape of long-range chromosomal interactions, suggesting a mechanism for long-range expression QTLs. In sum, our results show that molecular QTL mapping integrated across diverse populations provides a high-resolution view of how worldwide human genetic variation affects chromatin accessibility, gene expression, and phenotype.
**Editorial note:** This article has been through an editorial process in which the authors decide how to respond to the issues raised during peer review. The Reviewing Editor's assessment is that minor issues remain unresolved (see decision letter).
DOI: https://doi.org/10.7554/eLife.39595.001

**\*For correspondence:**
hbfraser@stanford.edu

**Competing interests:** The authors declare that no competing interests exist.

## Introduction

GWAS have been instrumental in advancing our understanding of complex traits. Indeed, these studies have successfully mapped thousands of loci associated with hundreds of human diseases and other traits (*Eicher et al., 2015*). However, the vast majority of causal variants that drive the associations remain unknown, since GWAS resolution is limited by the correlations between nearby variants known as linkage disequilibrium (LD). Pinpointing causal variants is essential not only for accurately predicting disease risk, but also for understanding the molecular mechanisms that underlie complex trait variation. Increasing GWAS sample size or genotyping density can improve resolution, but even the largest studies have yielded only a few dozen causal variants (*Farh et al., 2015*; *Huang et al., 2017*).

To achieve high mapping resolution, an alternative to increasing sample size within a single population is trans-ethnic fine-mapping, in which a GWAS is performed across multiple populations (*van de Bunt et al., 2015*; *Asimit et al., 2016*). Shared causal variants should be consistently associated with a trait across populations, while tag SNPs—those associated only because of their LD with a causal variant—may only be associated in a subset of populations, due to differing LD structures. This approach is especially effective when combining populations with disparate LD patterns, such as Europeans and Africans (*Asimit et al., 2016*). However, most GWAS have been restricted to European cohorts, limiting not only their mapping resolution, but also their ability to predict disease risk in non-European individuals (*Hindorff et al., 2018*).

Another approach to understanding molecular mechanisms underlying complex traits has been to intersect GWAS hits with quantitative trait loci (QTLs) for molecular-level traits, which provide an effective 'stepping stone' on the path from genotype to disease. QTLs involved in cis-regulation are especially informative, since the vast majority of GWAS hits are in noncoding regions (*Hindorff et al., 2009*). For example, QTLs for mRNA levels (eQTLs) are enriched for SNPs implicated by GWAS in a wide range of diseases, suggesting specific genes as likely mediators of the associations (*Lappalainen et al., 2013*; *Battle et al., 2017*). Similar enrichments have been reported for QTLs affecting mRNA splicing (*Fraser and Xie, 2009*; *Li et al., 2016*), transcription factor binding (*Waszak et al., 2015*; *Tehranchi et al., 2016*), and chromatin accessibility (*Kumasaka et al., 2016*; *Gate et al., 2018*). However, nearly all of these molecular-level QTL studies have been limited to a single population (most often either Europeans or Yorubans), thus limiting their utility for understanding worldwide human diversity—similar to the bias of GWAS for Europeans. Yet there is a great deal we could learn about the genetic basis of phenotypic diversity from performing trans-ethnic fine-mapping on molecular-level QTLs.

We recently developed an efficient approach for mapping molecular QTLs in which samples are pooled prior to sequencing. This pooling minimizes experimental variability between samples (both within and between experimental 'batches'), reducing the cost and effort of QTL mapping by over 25-fold compared to standard unpooled approaches (*Tehranchi et al., 2016*). However, our initial study suffered from the same limitations as most other molecular QTL studies: it involved a small cohort (60 individuals) from a single population. We reasoned that the efficiency of pooling could enable us to map *cis*-regulatory QTLs with far greater numbers of individuals and populations than would otherwise be possible, allowing us to perform trans-ethnic fine-mapping to pinpoint thousands of likely causal variants affecting *cis*-regulation and, in many cases, complex disease risk.

## Results

### Pooled QTL mapping of variants affecting chromatin accessibility

We applied pooled QTL mapping (*Kaplow et al., 2015*; *Tehranchi et al., 2016*) to chromatin accessibility (CA), a reliable indicator of local *cis*-regulatory activity, as measured by the Assay for Transposase-Accessible Chromatin (ATAC-seq) (*Buenrostro et al., 2015*). In this approach, many samples are combined into a single pool, in which ATAC-seq (or another assay of interest) is performed only once. Genetic variants that affect CA in cis will cause the more accessible allele to increase in frequency after ATAC compared to before, whereas variants with no effect on accessibility will have no significant change in frequency (*Figure 1A*). For each SNP, we estimated the post-ATAC allele frequency from read counts of each allele present in the ATAC-seq reads and used a regression-based approach to estimate pre-ATAC allele frequency; the significance of the difference between these two frequencies is our caQTL p-value (*Tehranchi et al., 2016*).

We selected a total of 1000 lymphoblastoid cell lines (LCLs) with full genome sequences (*Auton et al., 2015*) from 10 diverse populations: four European, four African, one African-American, and one Han Chinese (*Figure 1A*, *Figure 1—source data 1*). We combined 66–112 unrelated individuals from each population into one pool per population, and performed ATAC-seq with two biological replicates per population (*Figure 1—figure supplement 1*), resulting in a total of ~808 million informative autosomal reads from 20 pools. All populations showed a high correlation between pre-ATAC and post-ATAC allele frequencies (average Pearson r = 0.92; *Figure 1A*, *Figure 1—figure supplement 2*), as expected if most SNPs do not affect CA and thus have similar pre- and post-ATAC frequencies (*Tehranchi et al., 2016*). At a nominal $p<5\times10^{-4}$—corresponding to an irreproducible discovery rate (IDR, analogous to a false discovery rate; *Figure 1—figure supplement 3*) of ~1%—we mapped 13,657 to 26,182 independent caQTLs per population. This comprised 1.5–3.0% of all testable SNPs, defined as those with >2% minor allele frequency (MAF) and covered by at least 20 reads (*Figure 1B*, *Figure 1—figure supplement 4*). A total of 126,773 caQTLs were significant in at least one population (*Supplementary file 1*).

### Accuracy and efficiency of pooled QTL mapping

To gauge the accuracy of our caQTLs, we compared them with previously mapped QTLs. DNase-seq is another method used to assay chromatin accessibility, and SNPs affecting DNase read

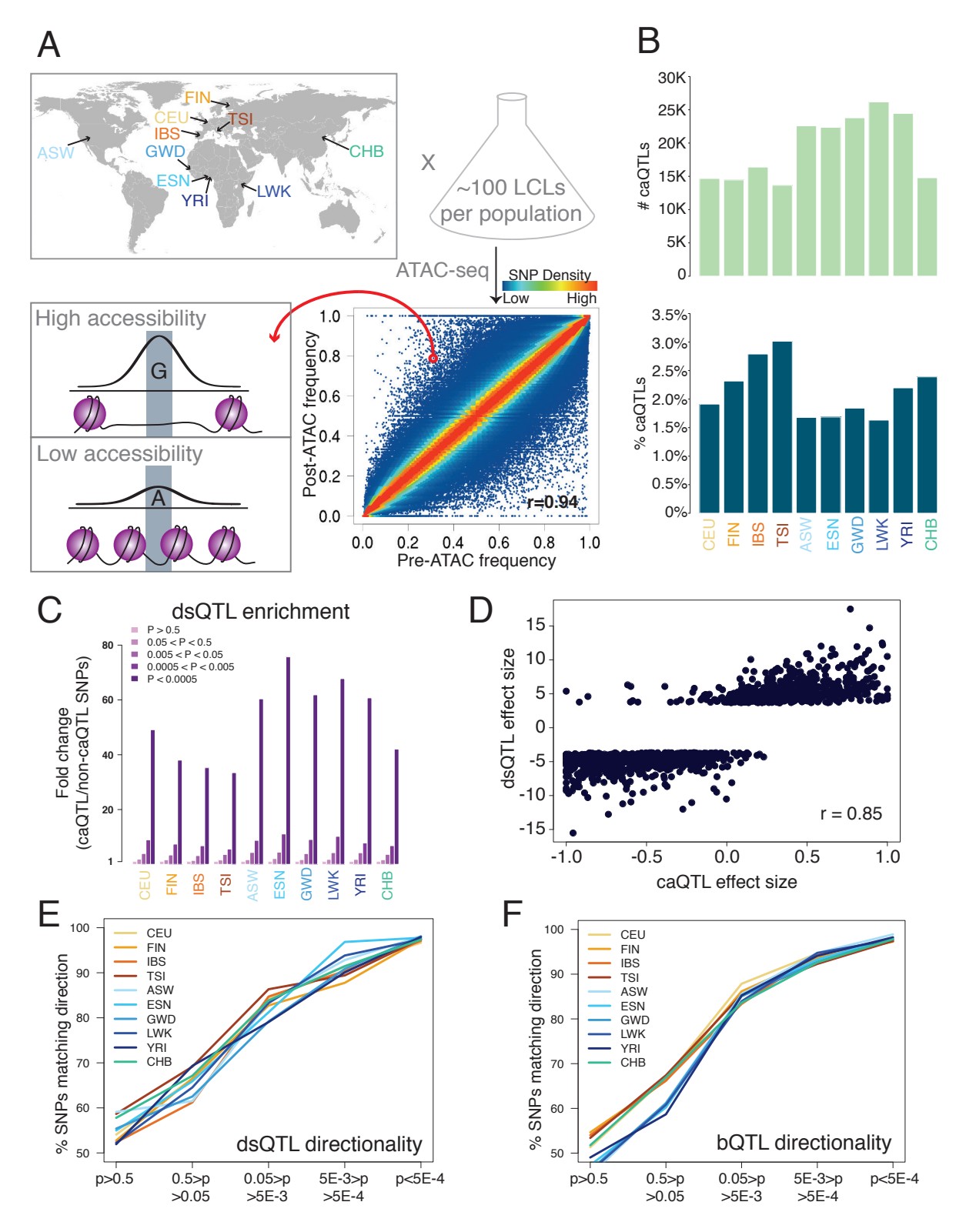

**Figure 1.** Outline and results of pooled ATAC-seq. (**A**) Performing ATAC-seq in a pool of individuals selects DNA molecules with higher CA, thus enriching for more accessible alleles. In this example (ASW population), the G allele has a low pre-ATAC frequency but a high post-ATAC frequency, due to its increased CA. The ten population abbreviations refer to: CEU, Utah residents with North European ancestry; FIN, Finnish; TSI, Tuscan; IBS, Iberian; ASW, African-American from Southwest US; YRI, Yoruban; ESN, Esan; LWK, Luhya; GWD, Gambian; and CHB, Han Chinese. (**B**) The number of

*Figure 1 continued on next page*

*Figure 1 continued*

caQTLs (top), and the percent of all tested SNPs called as caQTLs (bottom). (C) Enrichment of caQTLs among dsQTLs (*Degner et al., 2012*), at a range of caQTL p-value cutoffs. (D) Quantitative effect sizes of caQTLs and dsQTLs are highly correlated (scales of each axis are not comparable, and do not affect the correlation coefficient).( E–F) The degree of allelic concordance between our caQTLs and: (E) dsQTLs (*Degner et al., 2012*). (F) bQTLs aggregated for five TFs (*Tehranchi et al., 2016*). Full results available in *Figure 1—source data 1*.
DOI: https://doi.org/10.7554/eLife.39595.002

The following source data and figure supplements are available for figure 1:

**Source data 1.** Detailed results of analyses shown in the Figures.
DOI: https://doi.org/10.7554/eLife.39595.008

**Figure supplement 1.** Comparison of post-ATAC reference allele frequencies between biological replicates of each population pool.
DOI: https://doi.org/10.7554/eLife.39595.003

**Figure supplement 2.** Pre-ATAC vs post-ATAC reference allele frequencies for nine populations, similar to ASW plot in *Figure 1A*.
DOI: https://doi.org/10.7554/eLife.39595.004

**Figure supplement 3.** Top row: caQTL p-values for SNPs on chr one in ASW and CEU, shown separately for each biological replicate.
DOI: https://doi.org/10.7554/eLife.39595.005

**Figure supplement 4.** QQ plots of expected (under the null) vs observed caQTLs p-values.
DOI: https://doi.org/10.7554/eLife.39595.006

**Figure supplement 5.** caQTLs enrichments among other molecular QTLs (*Ding et al., 2014*, *Lappalainen et al. (2013)*, *Waszak et al., 2015*, *Tewhey et al., 2016*, *Banovich et al., 2014*).
DOI: https://doi.org/10.7554/eLife.39595.007

density—known as dsQTLs—have been mapped using 70 individual (non-pooled) Yoruban LCLs (*Degner et al., 2012*). We compared our caQTLs to dsQTLs in three ways. First, testing the overlap between these two sets, we found a 33- to 76-fold enrichment of dsQTLs among caQTLs, with highest enrichment in the five African/African-American populations (*Figure 1C*; enrichments for other QTLs in *Figure 1—figure supplement 5*). Second, we compared the quantitative effect sizes of caQTLs and dsQTLs, and found excellent agreement (Pearson $r = 0.85$; *Figure 1D*). Third, we tested how the directionality agreement (whether caQTLs and dsQTLs call the same allele as more accessible) changes with the caQTL p-value, and found that agreement increased with more stringent cutoffs, reaching 96.7–98.1% agreement at caQTL $p < 5 \times 10^{-4}$ (*Figure 1E*). These three tests show that our caQTLs are consistent with dsQTLs mapped with unpooled samples.

Because accessible chromatin is more often bound by TFs, we also tested the directionality agreement between overlapping caQTLs and TF binding QTLs (bQTLs) aggregated for five TFs (*Tehranchi et al., 2016*), since we expect increased TF binding to be associated with increased CA. Once again, we found that the agreement increases with more stringent p-value cutoffs, reaching 97.3–98.9% agreement at caQTL $p < 5 \times 10^{-4}$ (*Figure 1F*). This high level of agreement suggests a low rate of false positive caQTLs, consistent with our estimated 1% IDR.

To assess the efficiency of our pooling approach, we compared our results with caQTLs mapped with RASQUAL—a computational QTL mapping approach that accounts for many possible confounding variables—in non-pooled European LCLs (*Kumasaka et al., 2016*). At a matched cutoff in our CEU population, we mapped 2.8-fold more caQTLs with 15-fold fewer reads and 12-fold fewer ATAC-seq libraries—a ~ 40 fold improvement in cost per caQTL (see Supplemental Note). Overall, our comparisons with previously mapped QTLs (*Figure 1C–F*, *Figure 1—figure supplement 5*) suggest that pooling is an efficient strategy that agrees well with unpooled QTL mapping.

## caQTLs shared across populations

We next assessed the extent of caQTL sharing across populations. A caQTL might not be shared due to biological causes, such as dependence of a variant's effect on the genetic background (epistasis), or due to trivial causes such as not meeting our 20 read cutoff in one population. To exclude trivial cases we focused on 142,049 variants that were testable in all ten populations. Among these, we observed a clear trend for increased sharing within continents: the mean fraction of shared caQTLs was 59.9% within Africans and 59.8% within Europeans, compared to 48.4% between these two groups (*Figure 2A*). The Han Chinese (CHB) caQTLs were shared moderately with Europeans (52.4%) and less well with Africans (47.7%), reflecting their closer relatedness to Europeans. In addition, African-American (ASW) caQTLs showed greater sharing with all five European/Han populations

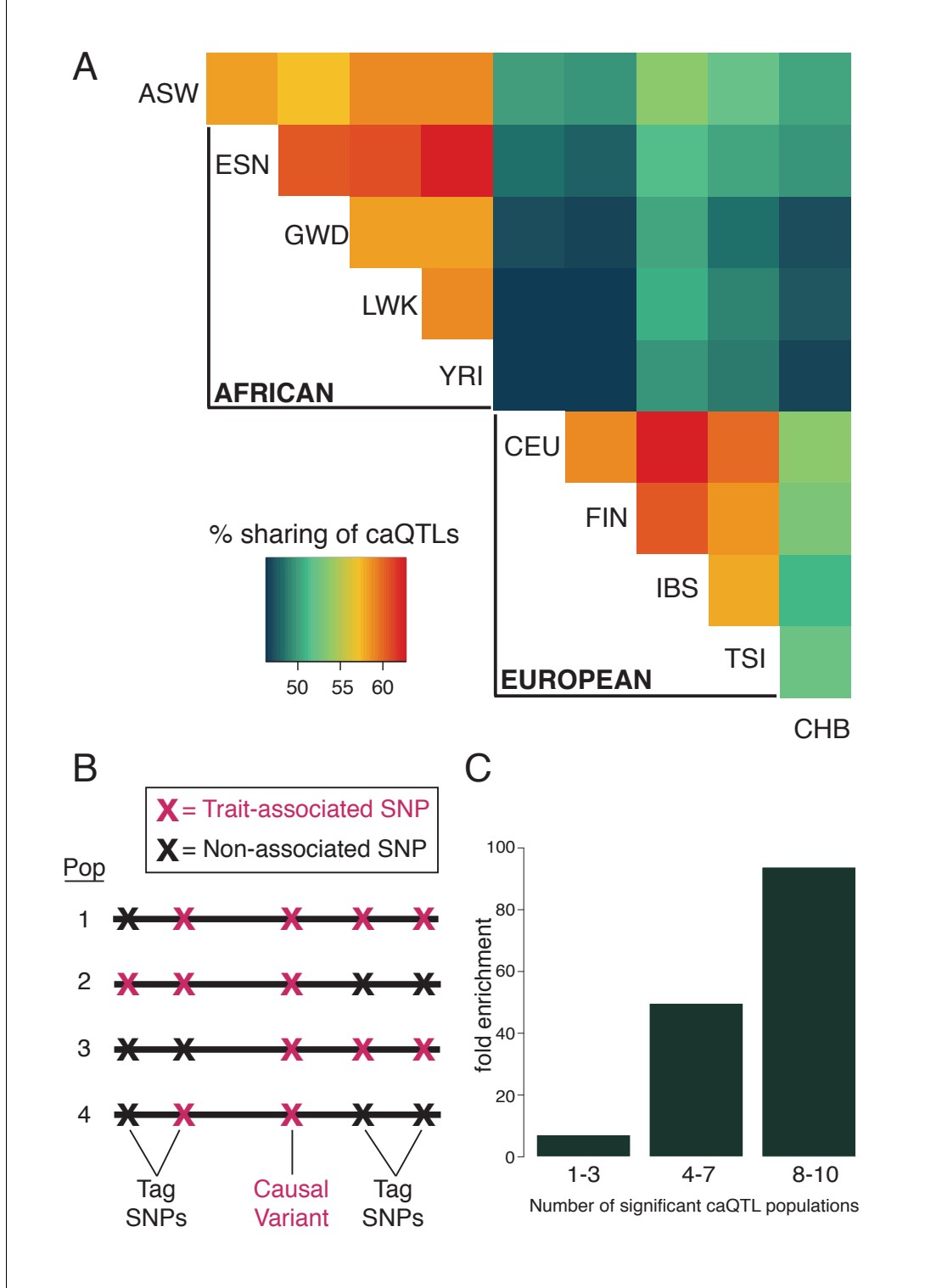

**Figure 2.** Fine-mapping shared caQTLs. (**A**) Heatmap showing the overlap in caQTLs for every pair of populations (only for variants that were testable in all ten). To avoid issues related to arbitrary p-value cutoffs, we used the shift in p-value distribution, known as $\pi_1$ (*Storey et al., 2004*), to assess overlap. (**B**) Mapping a trait in multiple populations differing in LD structure allows fine-mapping of causal variants, which will show the most consistent associations. (**C**) caQTLs shared across many populations (at $p<5\times10^{-4}$) are more highly enriched for experimentally-determined causal eQTL variants (*Tewhey et al., 2016*). Full results available in *Figure 2—source data 1*.

DOI: https://doi.org/10.7554/eLife.39595.009

*Figure 2 continued on next page*

*Figure 2 continued*

The following source data and figure supplements are available for figure 2:

**Source data 1.** All caQTLs testable (at least 20 reads and MAF >0.01) in all ten populations.
DOI: https://doi.org/10.7554/eLife.39595.012
**Figure supplement 1.** Sharing of caQTLs across populations, as in *Figure 2A*, but excluding comparisons with divergent allele frequencies.
DOI: https://doi.org/10.7554/eLife.39595.010
**Figure supplement 2.** Example of a shared caQTL (rs79979970) that is individually significant in only one population (CHB) out of eight tested, but reaches a shared caQTL p=$5.6\times10^{-7}$ because it has p<0.1 in an additional four populations.
DOI: https://doi.org/10.7554/eLife.39595.011

than any of the four African populations did, consistent with their admixed ancestry. Restricting the analysis to caQTLs with similar allele frequencies across populations led to a similar pattern (*Figure 2—figure supplement 1*). The concordance between caQTL sharing and known phylogenetic relationships suggests that some caQTLs may have population-specific effects, as has also been observed for other types of molecular QTLs (*Stranger et al., 2012*; *Fraser et al., 2012*), though the biological mechanisms underlying this divergence will require further study.

A fundamental limitation of human GWAS is their inability to pinpoint most causal variants, due to linkage disequilibrium (LD) that often results in many variants with equally strong associations (*van de Bunt et al., 2015*). However, causal variants can be fine-mapped by combining association data across populations, especially with African populations due to their low LD (*Asimit et al., 2016*). Causal variants should be consistently associated with a trait across populations, while tag SNPs—those associated only because of their LD with a causal variant—may only be associated in a subset of populations, due to differing LD structures (*Figure 2B*).

We performed fine-mapping by searching for caQTLs present in multiple populations. To test whether this was indeed enriching for causal variants, we intersected our caQTLs with a collection of experimentally verified causal eQTL variants (*Tewhey et al., 2016*). We found increasing enrichment for known causal variants with increasing number of significant caQTL populations (*Figure 2C*), suggesting that combining the diverse LD patterns improved our mapping resolution substantially. In order to account for both the number of significant populations as well as the caQTL strength within each population (*Figure 2—figure supplement 2*), we calculated Fisher's combined p-value for each caQTL across all ten populations; at a combined p<$5\times10^{-6}$, we identified 45,243 SNPs, which we refer to as 'shared caQTLs' (*Supplementary file 2*). Nearly all (99.8%) of these were significant across multiple populations, and 98.6% showed concordance of the more accessible allele across populations.

## Characterizing fine-mapped caQTLs

Leveraging the high resolution of our fine-mapped shared caQTLs, we first investigated their genomic locations. We found that they were most highly enriched near active enhancers and transcription start sites (TSSs) (*Ernst and Kellis, 2012*), accounting for 54% of shared caQTLs in just 3.1% of the genome (*Figure 3A*, purple and dark green). However, these enrichments were primarily driven by ATAC-seq read density, reflecting greater chromatin accessibility in these regions; after controlling for read density, we found the strongest enrichments in weak enhancers (2.3-fold) and quiescent regions (3.6-fold), and 8.7-fold depletion near TSSs. We hypothesize that mutations affecting accessibility near TSSs are more likely to be deleterious, and thus selected against, resulting in the observed depletion. This 8.7-fold depletion near TSSs is greater than the analogous depletion of nonsynonymous changes in exons; in fact, we estimated that that these caQTLs are ~81% more likely to be deleterious, and removed by selection, than nonsynonymous mutations (see Supplemental Note). Consistent with this, we also found that caQTLs near active TSSs have lower minor allele frequencies than elsewhere in the genome (median MAF = 0.17 for active TSS regions and 0.18 for flanking active TSS regions, compared to 0.21 for caQTLs elsewhere; Wilcoxon p=$1.5\times10^{-28}$ and $8\times10^{-6}$ respectively).

Disruption of TF binding motifs can lead to caQTLs (*Degner et al., 2012*; *Kumasaka et al., 2016*). To investigate the effects of caQTL variants on TF binding motifs we searched for motifs enriched specifically in the more accessible caQTL alleles (by using the less accessible alleles as the background set), or in the less accessible alleles (see Materials and methods). We found 80 known

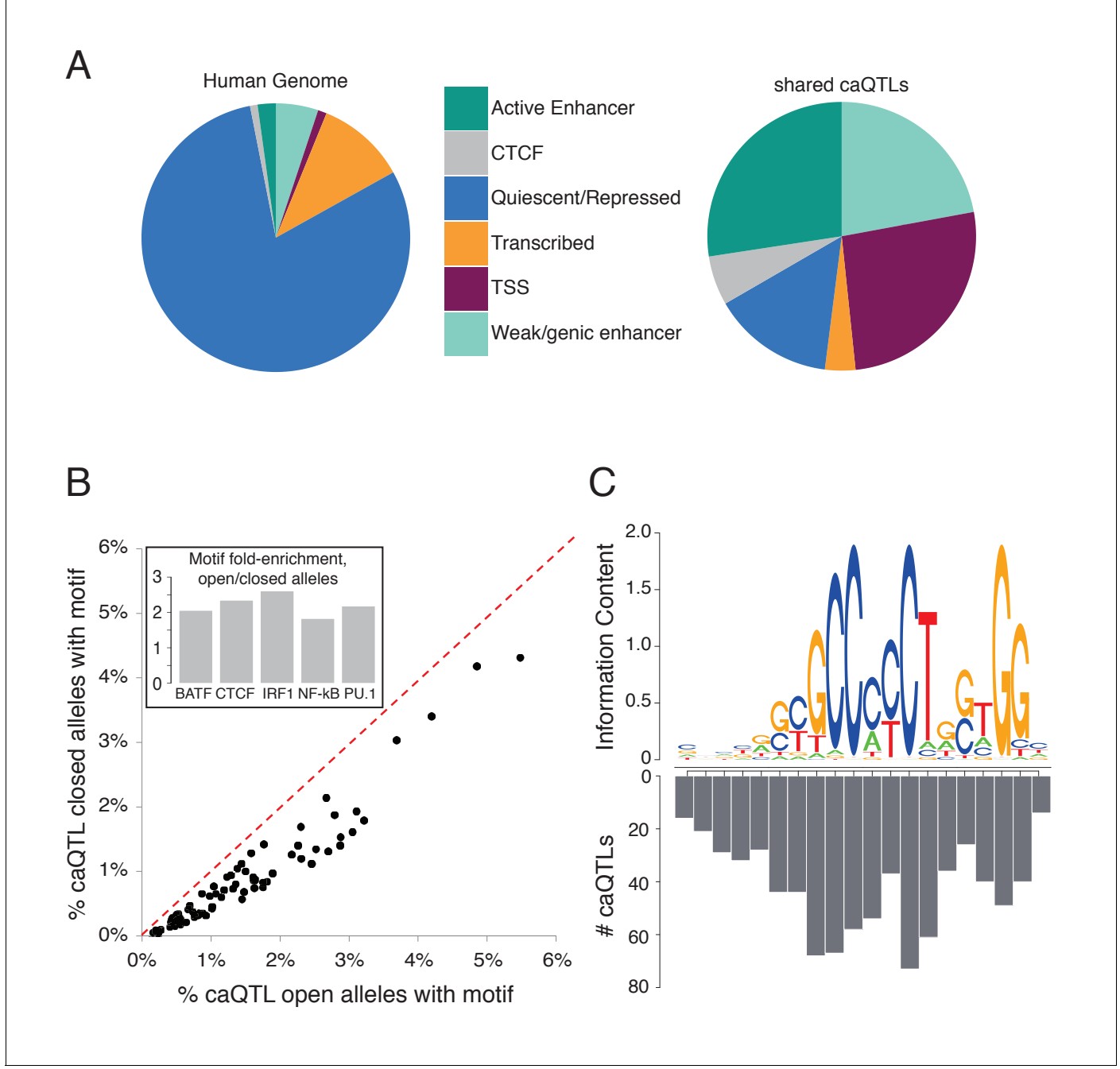

**Figure 3.** Characterizing shared caQTLs. (**A**) The fraction of the genome (left) and of shared caQTLs (right) in each of four classes, annotated based on chromatin signatures (*Ernst and Kellis, 2012*). TSS includes TSS flanking regions; full results in *Supplementary file 2*. (**B**) Searching for motifs enriched specifically among open alleles (using closed alleles from the same caQTLs as the background comparison set), we found 80 motifs enriched among open alleles (points below the diagonal). Repeating the analysis for closed alleles, we found no motifs enriched (above diagonal). Note that many motifs are partially overlapping, and thus not independent. Inset: fold-enrichment in open/closed alleles for five selected TFs. Full results in *Figure 3— source data 1*. (**C**) The number of caQTLs overlapping each position within the CTCF motif strongly mirrors the information content (i.e. the importance for binding) of that position, as expected if these caQTLs are causal variants affecting CA via CTCF binding. Full results available in *Figure 3—source data 1*.

DOI: https://doi.org/10.7554/eLife.39595.013

The following source data and figure supplement are available for figure 3:

**Source data 1.** *Fig 3A*: Numbers in *Figure 3A*, and chromatin states for every caQTL .

DOI: https://doi.org/10.7554/eLife.39595.015

*Figure 3 continued on next page*

*Figure 3 continued*

**Figure supplement 1.** Effect of shared caQTLs on DNA shape.
DOI: https://doi.org/10.7554/eLife.39595.014

TF motifs enriched in the open allele sequences (FDR $\leq$ 0.1% for each; *Figure 3B*), but none enriched in the closed alleles from the same caQTL loci. This striking asymmetry supports the idea that a major mechanism leading to caQTLs is disruption of TF binding, where caQTL variants matching the consensus motif—and thus promoting TF binding—result in more accessible chromatin. The most highly enriched motifs included TFs specific to immune cells like BATF, as well as more ubiquitous factors like CTCF (*Figure 3B* inset).

To further investigate fine-mapped caQTLs in TF binding motifs, we focused on CTCF, due to its long and well-characterized motif (*Ding et al., 2014*). We reasoned that the most critical positions in the CTCF motif (i.e. those with the least variation across CTCF binding sites, represented by high information content in the position weight matrix) should be more likely to affect CTCF binding and chromatin when disrupted, and thus should be the most enriched for caQTLs (Ding et al. 2012, *Maurano et al., 2015*). Comparing shared caQTL density with the CTCF motif, we indeed found that the caQTL density is strongly correlated with the information content of each motif position (Pearson $r$ = 0.86; *Figure 3C*).

Since the shape of the DNA double helix is sequence-dependent, and can affect interactions with TFs (*Chiu et al., 2016*), we then tested whether shared caQTL variants tend to cause larger changes in DNA shape than non-caQTL variants matched for ATAC-seq read depth. We found small but significant differences in how caQTLs impact DNA shape (*Figure 3—figure supplement 1*) suggesting that DNA shape may contribute to variation in CA.

In sum, our analyses of the fine-mapped caQTLs support the idea that these are highly enriched for causal variants, since non-causal SNPs in LD with causal variants would not be expected to disrupt known motifs (*Figure 3B*) or critical motif positions (*Figure 3C*).

## Chromatin accessibility impacts long-range chromosomal interactions

CTCF has a well-established role in mediating long-range interactions between chromosomal loci, an essential component of transcriptional regulation (*Guo et al., 2015*; *Sanborn et al., 2015*; *Rao et al., 2014*). Consistent with this role, we previously reported that bQTL alleles increasing CTCF binding also increase these long-range interactions (*Tehranchi et al., 2016*). To test if caQTLs also affect long-range interactions, we measured how often the more accessible allele had significantly more long-range interactions (with loci > 20 kb away) (*Rao et al., 2014*) than the less accessible allele, and vice versa; we found a 2.5-fold enrichment of the more accessible allele having more interactions (*Figure 4A*; binomial p<10$^{-36}$), similar to the 2.2-fold bias of CTCF bQTLs (*Tehranchi et al., 2016*). The allelic ratio increased slightly (to 2.7-fold) when restricted to interactions > 100 kb apart. This enrichment was not observed for inter-chromosomal interactions (*Figure 4—figure supplement 1*), suggesting it is unlikely to be due to a nonspecific bias in the Hi-C assay. These results establish a role for CA in polymorphic long-range chromosomal interactions of a similar magnitude as CTCF.

Our previous work showed that in addition to CTCF bQTLs, bQTLs for five other TFs also affect long-range chromosomal interactions, suggesting a possible role for many TFs in chromosomal architecture (*Tehranchi et al., 2016*); however, the underlying mechanism and causality has not been investigated. We hypothesized that we could isolate the effects of TF binding and CA by separating bQTLs into two groups: those that affect TF binding and CA, and those that affect TF binding but not CA (*Figure 4—figure supplement 2*). Comparing the effects of these two groups on long-range interactions could then shed light on whether TFs can shape chromosomal interactions independent of CA, or if instead CA effects are essential for TFs to impact these interactions.

We found a strong and consistent effect on long-range interactions for bQTLs that affect accessibility, in which the allele with increased TF binding is biased towards having more interactions (*Figure 4B* left; binomial p<0.008 for each). In contrast, we found no significant effect for the bQTLs that do not affect accessibility, for all six TFs (*Figure 4B* right; binomial p>0.08 for each). This suggests that TF binding alone has no detectable association with long-range interactions; even for

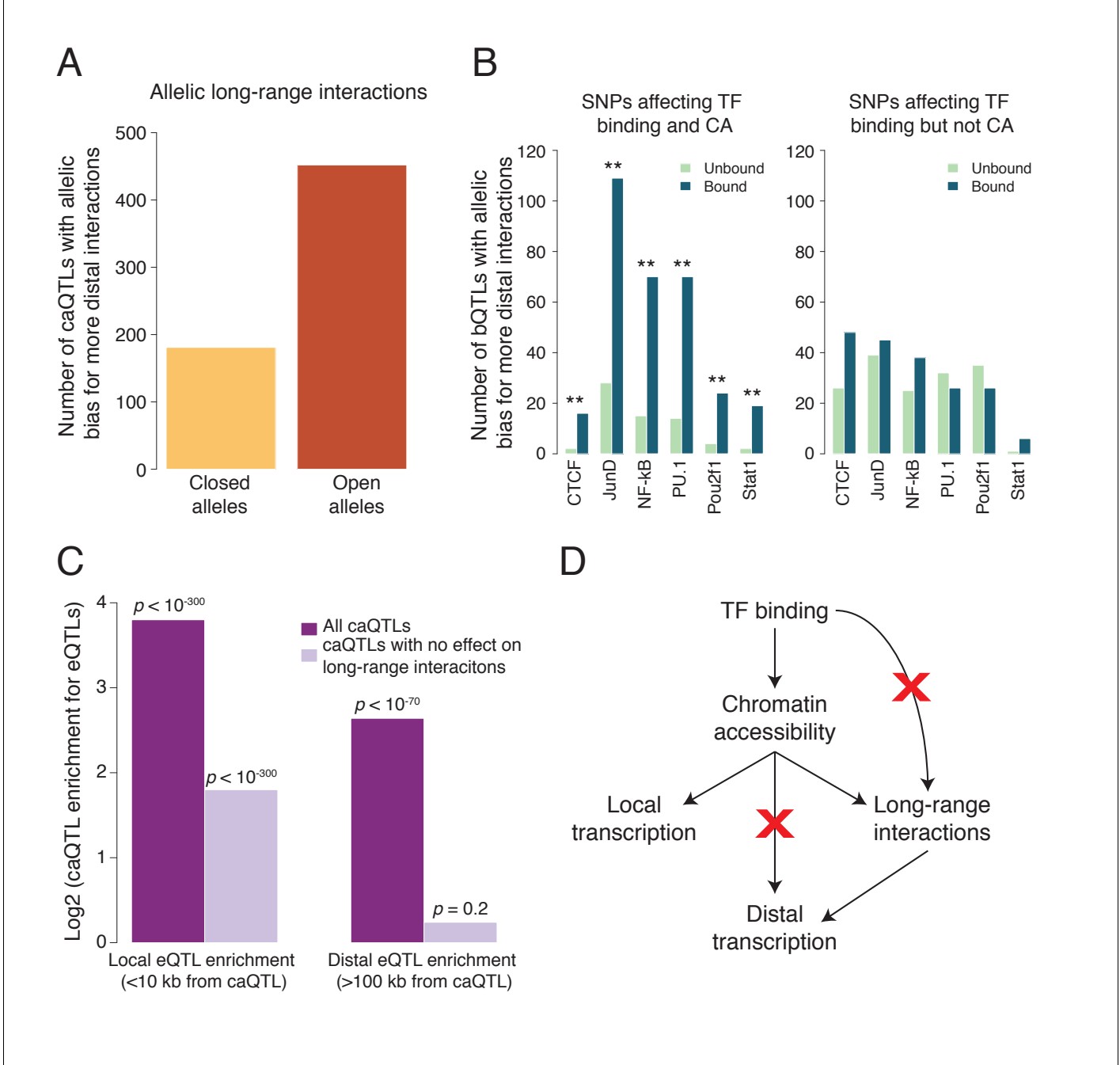

**Figure 4.** TF binding and chromatin accessibility. (A) Using allele-specific 3D chromosomal interaction (Hi-C) data from an LCL (*Rao et al., 2014*), we found that open alleles of caQTLs tend to have more long-range interactions than do the closed alleles, establishing a role for CA in polymorphic chromosomal interactions.(B) Splitting bQTLs into two groups (*Figure 4—figure supplement 2*), we found that bQTLs were strongly associated with the extent of long-range interactions only when they also affect CA (left panel; ** indicates Bonferroni-corrected binomial p<0.008 for all six TFs); for bQTLs that do not affect CA, no allelic bias was observed (right panel; Bonferroni-corrected binomial p>0.08 for all six TFs). (C) caQTLs are strongly enriched for both local and distal eQTLs; however among those that do not affect long-range chromosomal interactions, only local eQTLs are enriched. (D) Model summary: our results suggest that bQTLs generally cannot affect long-range chromosomal interactions without an effect on CA, and caQTLs generally cannot affect distal transcription without an effect on long-range interactions. The model shown represents a plausible interpretation, but is not the only possible causal scenario. Full results available in *Figure 4—source data 1*.

DOI: https://doi.org/10.7554/eLife.39595.016

The following source data and figure supplements are available for figure 4:

**Source data 1.** *Fig 4A*: Numbers going into *Figure 4A*, including additional distance cutoffs.

*Figure 4 continued on next page*

*Figure 4 continued*

DOI: https://doi.org/10.7554/eLife.39595.021

**Figure supplement 1.** Allelic bias of shared caQTLs for inter-chromosomal interactions.

DOI: https://doi.org/10.7554/eLife.39595.017

**Figure supplement 2.** Venn diagram indicating three possible combinations of caQTL/bQTL overlaps, and how we used these to infer their downstream effects in *Figure 4B*.

DOI: https://doi.org/10.7554/eLife.39595.018

**Figure supplement 3.** Causal probabilities of SNPs affecting disease risk (*Farh et al., 2015*) for two examples discussed in the main text.

DOI: https://doi.org/10.7554/eLife.39595.019

**Figure supplement 4.** Likelihood of caQTLs from LCLs acting as eQTLs in other tissues.

DOI: https://doi.org/10.7554/eLife.39595.020

CTCF, an effect on CA is also necessary. Together with additional evidence (See Supplemental Note), we propose that CA is a key intermediary between TF binding and chromosomal interactions (*Figure 4D*).

## caQTLs that regulate distal genes alter the landscape of long-range interactions

We then asked whether these polymorphic chromosomal interactions are involved in the regulation of transcription by distal caQTLs (defined as a caQTL that is also an eQTL for a gene whose TSS is >100 kb away). Enhancers can act through physical interactions with promoters, and genetic variants can impact chromatin and transcription at distal loci with which they physically interact (*Waszak et al., 2015*, *Grubert et al., 2015*, *Tehranchi et al., 2016*). However, whether these long-range effects of SNPs depend on changes in the patterns of chromosomal interactions—as opposed to being mediated by static interactions—has not been investigated.

To address this question, we controlled for the effect of changes in long-range interactions by analyzing caQTLs with no measurable effect on these interactions (see Methods). If caQTL effects on distal genes do not depend on changes in these interactions, then eQTL enrichment should not be affected by controlling for them; if instead changes in these interactions are necessary for long-range regulation, then we would expect to see a sharp reduction in eQTL enrichment when controlling for them.

As expected, we found that overall caQTLs were strongly enriched for local eQTLs (TSS <10 kb from the caQTL); restricting the analysis to caQTLs with no effect on interactions reduced the magnitude of this enrichment, but it was still highly significant in both cases (*Figure 4C* left; Fisher's exact $p<10^{-300}$ for each enrichment), suggesting that changes in long-range interactions are not necessary for caQTLs to affect transcription of nearby genes. caQTLs were also enriched for distal eQTLs ($p<10^{-70}$), but this enrichment was entirely lost when restricting the analysis to caQTLs with no change in long-range interactions (*Figure 4C* right; p=0.20). This suggests that alterations in the patterns of long-range chromosomal interactions are necessary for caQTLs to affect distal gene expression (*Figure 4D*), and therefore that these polymorphic interactions may be an important mechanism by which eQTLs can act over vast genomic distances.

## Fine-mapping GWAS associations with caQTLs

In addition to revealing insights into transcriptional regulation, caQTLs also provide a means to explore genotype/phenotype associations by identifying likely causal variants and their molecular mechanisms of action. To compare our fine-mapping approach with standard GWAS fine-mapping, we asked whether our shared caQTLs were more likely to be assigned a high probability of being causal for disease risk (*Farh et al., 2015*). We found that for several autoimmune diseases—the class of disease most directly relevant to LCLs—shared caQTLs were highly enriched for likely causal variants (e.g. for Crohn's disease, mean causal probability for shared caQTLs was 2.4-fold higher than for non-caQTLs, p=0.002; for ankylosing spondylitis, mean causal probability for shared caQTLs was 2.6-fold higher, p=0.001). However for most diseases, the number of overlaps between these two data sets was too small to conduct a meaningful test. Together with other evidence of the efficacy of our fine-mapping approach (*Figure 2C*), this suggests that our shared caQTLs are an effective means of resolving likely causal cis-regulatory variants.

We illustrate different types of insights with three examples from autoimmune diseases. In rare cases, a lack of LD allows GWAS to implicate a single likely causal variant; this is the case for rs4409785, which is associated with rheumatoid arthritis, multiple sclerosis, and vitiligo (*Sawcer et al., 2011*; *Jin et al., 2012*, *Okada et al., 2014*), and was predicted to be the causal variant for vitiligo with 85% probability (*Farh et al., 2015*) (*Figure 4—figure supplement 3*). The SNP is bound by CTCF in LCLs (*ENCODE Project Consortium, 2012*) and disrupts a CTCF motif; we found that it is a caQTL in two European populations (FIN and TSI, $p<3\times10^{-6}$ in each; not tested in African or ASW populations due to its low MAF), thus providing evidence of an effect on accessibility that is likely mediated by CTCF.

In other cases, multiple SNPs in LD have conflicting evidence of causality. For example, rs479844 is a lead SNP associated with atopic dermatitis in Europeans, and although it was assigned a 90% probability of being causal (*Farh et al., 2015*) (*Figure 4—figure supplement 3*), it has not replicated in some populations (*Lepre et al., 2013*). A meta-analysis reported that when non-European ethnicities are included, the only consistent association is for another nearby SNP, rs10791824 (*Paternoster et al., 2015*), which is in strong LD with rs479844 in Europeans ($r^2 = 0.92$) but not in our four African populations ($r^2 = 0.21$). This variant is a strong caQTL in all six non-European populations ($p<10^{-42}$ in each) and also significant in 2/4 European populations ($p<5\times10^{-4}$ in each), supporting the trans-ethnic GWAS (*Paternoster et al., 2015*) implicating it as the likely causal variant. In addition, the caQTL is an eQTL for *OVOL1* (*Lappalainen et al., 2013*; *Battle et al., 2017*), a TF involved in skin development, and has been found to affect transcription in a large-scale reporter assay in LCLs (*Tewhey et al., 2016*).

The third and most frequent case is where many SNPs are in strong LD, and thus have almost equally strong associations with a disease. For example, a large LD block of 62 variants on chromosome three is associated with multiple sclerosis, all of which have <5% probability of being causal based on GWAS signal alone (*Farh et al., 2015*). However one of these, rs485789, is a caQTL shared across 8/10 populations ($p<2\times10^{-5}$ in each). Interestingly, this was also predicted as a causal variant by a sequence-based predictor of regulatory variants (*Lee et al., 2015*), and is an eQTL for *IL12A* (*Lappalainen et al., 2013*), a cytokine implicated in several autoimmune diseases (*Guo et al., 2016*). Therefore, rs485789 is a likely causal variant for multiple sclerosis that acts on *IL12A* via its effect on CA.

More broadly, we found 5598 caQTLs that were also associated with disease risk or other complex traits (GWAS $p<5\times10^{-8}$; *Supplementary file 3*). Although most GWAS loci include dozens of potential causal variants in LD, there are only ~2.2 caQTLs per GWAS locus, providing a far smaller credible set for targeted follow-up studies. Among these, 115 caQTLs were shared across all ten populations, suggesting that many of these are likely to be causal for disease risk; we highlight ten examples in *Table 1*. We note that although our caQTLs were measured in LCLs, the traits they are associated with are related to a wide range of tissues. Consistent with this, we found that the regulatory effects of caQTLs are typically shared across most tissues, suggesting that their effects on CA are broadly shared as well (Supplemental Note, *Figure 4—figure supplement 4*). Therefore, it should not be surprising that these caQTLs can contribute to risk for diseases that have no clear connection to LCLs.

## Discussion

By applying pooled QTL mapping in 1000 cell lines from 10 diverse populations, we achieved unprecedented power and resolution to fine-map *cis*-regulatory variants. The resulting collection of over 100,000 caQTLs allowed us to refine our understanding of the sequence-level causes, as well as the phenotypic consequences, of natural variation in chromatin accessibility.

More generally, our results show that molecular QTL mapping and trans-ethnic fine-mapping are complementary in several respects. Molecular QTL mapping provides information about molecular mechanisms and allelic effect direction, while including multiple populations allows for fine-mapping of causal variants; combined, these can provide unprecedented insights into the genetic basis of human variation. Moreover, since variation in molecular-level traits such as CA can be mapped to over $10^5$ QTLs, they provide tremendous opportunities for fine-mapping regulatory variants throughout the genome, which can then inform follow-up studies for diseases or other traits where trans-ethnic fine-mapping may not be feasible (e.g. *Table 1*). We expect that future studies will greatly

**Table 1.** Ten candidate causal variants, shared as caQTLs across all 10 populations.
GWAS information is from the GRASP database (*Eicher et al., 2015*). See *Supplementary file 3* for all caQTL/GWAS overlaps.

| Chr | caQTL position | caQTL rsID | GWAS rsID | GWAS p-value | GWAS trait | Candidate gene |
|---|---|---|---|---|---|---|
| 15 | 86018746 | rs7161880 | rs4281668 | 9.90E-09 | Aggressive prostate cancer | *AKAP13* |
| 9 | 5097544 | rs3780372 | rs10974944 | 5.10E-32 | Myeloproliferative neoplasms | *JAK2* |
| 12 | 111351439 | rs11065769 | rs10849917 | 3.21E-08 | Coronary artery disease | *MYL2* |
| 2 | 68598955 | rs17035378 | rs17035378 | 8.00E-09 | Celiac disease | *PLEK* |
| 6 | 167433948 | rs6904946 | rs2301436 | 1.00E-12 | Crohn's disease | *RNASET2* |
| 14 | 69273905 | rs194749 | rs194749 | 2.70E-10 | Inflammatory bowel disease | *ZFP36L1* |
| 1 | 154359411 | rs9651053 | rs11265608 | 2.75E-08 | Juvenile idiopathic arthritis | *IL6R* |
| 4 | 1731653 | rs798764 | rs798766 | 3.90E-13 | Urinary bladder cancer | *FGFR3* |
| 13 | 40334852 | rs9603612 | rs9532434 | 4.52E-08 | Juvenile idiopathic arthritis | *COG6* |
| 11 | 118560857 | rs73001406 | rs11216930 | 1.40E-09 | Glioma | *PHLDB1* |

DOI: https://doi.org/10.7554/eLife.39595.022

benefit from this combination, and that the ease of pooled QTL mapping will facilitate the adoption of this integrated approach.

As a first step towards understanding how caQTLs impact phenotypes, we have explored their effects on local and distal transcriptional regulation. This strategy is conceptually similar to experiments involving genome editing of specific regulatory elements (*Guo et al., 2015*; *Sanborn et al., 2015*), except that we are utilizing natural genetic variation as our source of perturbations; this allows us to examine many more variants, though we are restricted to those that are not strongly deleterious (since these are too rare for us to observe). Our results suggest that TF binding affects long-range chromosomal interactions via its effect on CA, and that CA affects transcription of distal genes via its effect on long-range chromosomal interactions (*Figure 4*). This latter inference is especially intriguing, since it suggests a molecular mechanism underlying distal eQTLs: variants alter enhancer activities, which in turn alter the physical interactions of enhancers with distal genes. To our knowledge, this represents the first evidence that variation in the landscape of chromosomal interactions underlies long-range genetic effects on transcription.

There are several important limitations to our study. First, we have yet to map the majority of human caQTL variants—for example those specific to other cell types, those that affect CA of distal loci (in trans or long-range *cis*), or those involving rare variants or small effect sizes. Second, our shared caQTLs should be regarded only as candidate causal variants to guide future experiments, similar to other statistically fine-mapped associations. Third, our fine-mapping approach is distinct from typical GWAS fine-mapping where many variants in a locus are genotyped in a large cohort; our pooled approach can only map caQTLs that are covered by ATAC-seq reads. Within these regions of open chromatin, our approach genotypes essentially all common variants, but variants outside these regions are not covered. Despite this limitation, our increased efficiency resulted in over 10-fold more caQTLs than previous studies (*Degner et al., 2012*; *Kumasaka et al., 2016*). Fourth, like all forms of trans-ethnic fine-mapping, our method will only work when a variant is present at an appreciable frequency, and affects the same trait, across multiple populations. And finally, much work remains to understand how the caQTLs we have mapped affect human phenotypes.

Looking ahead, we expect that combining GWAS with pooled QTLs mapped in diverse populations and cell types will be an efficient and highly effective strategy for identifying causal variants and their molecular mechanisms of action, two major bottlenecks in bridging the gap between GWAS and disease etiologies.

## Materials and methods

### Cell culture conditions

Lymphoblastoid cell lines (LCLs) from unrelated individuals were obtained from the Coriell Institute (http://www.coriell.org). No authentication of cell line identity or mycoplasma contamination was performed. The LCLs were grown in RPMI-GlutaMAX-HEPES media (Life Technologies, 72400) with 15% FBS, 100 I.U./mL penicillin, and 100 µg/mL streptomycin at 37°C, 5% $CO_2$.

### ATAC-seq

We used a modified version of the ATAC-seq protocol (*Buenrostro et al., 2015*). Each individual cell line was grown to a density of $6-8 \times 10^5$ cells/mL and $2 \times 10^3$ cells were collected and pooled by population such that each individual was approximately equally represented in the pool. Sub-pools were frozen in liquid nitrogen at $-180°C$. After all 1000 individuals were collected, subpools were combined by population and fresh media was added up to 15 ml, centrifuged at 1500 rpm for 10 min, and supernatant removed. The pellet was resuspended in 0.75 ml media with 200 units/mL DNase and incubated in a ThermoMixer for 30 min at 37°C at 300 rpm. 0.8 mL of Ficoll-paque Plus (GE Cat #17-1440-03) was added to a 2 mL tube and the 0.75 mL cell suspension was carefully layered on top. Samples were centrifuged at 500x *g* for 20 min at room temperature with no brake. The thin cloud of live cells in the middle layer were pipetted to a new tube, washed in 1 ml 1x PBS buffer, and split into two tubes with $10^5$ cells for replicates. Tubes were centrifuged at $500 \times g$ for 5 min at 4°C and supernatant was removed.

ATAC-seq was performed simultaneously on all 20 replicates. To each cell pellet, 100 ul of transposition mix was added (50 uL 2x TD Buffer (Illumina Cat #FC-121–1030), 5.0 uL Tn5 Transposase (Illumina Cat #FC-121–1030), 42 uL nuclease-free water, 1 uL 10% Tween-20, 3 uL 1% Digitonin). Samples were incubated in a ThermoMixer at 37°C for 30 min @ 750 rpm, then purified using Qiagen MinElute Kit with DNA eluted in 11 uL 10 mM Tris buffer, pH 8. Transposed DNA fragments were amplified using PCR where total cycles were calculated using qPCR as described in (*Buenrostro et al., 2015*). Amplified libraries were purified using Qiagen PCR Cleanup Kit and eluted in 21 uL 10 mM Tris pH 8. An additional purification step was performed using a 1:1.2 ratio of DNA:AMPure XP beads. Libraries were sequenced on an Illumina HiSeq 4000 (150 bp, paired-end reads).

### Mapping ATAC-seq reads

To remove adapters, reads were trimmed using cutadapt (*Martin, 2011*) with the following command: cutadapt -e 0.20 -a CTGTCTCTTATACACATCT -A CTGTCTCTTATACACATCT -m 5 -o fastq1-out -p fastq2out fastq1 fastq2

Trimmed reads were mapped using a modified version of the WASP pipeline for controlling mapping bias (*van de Geijn et al., 2015*) with scripts find_intersecting_snps_2.py and filter_remapped_reads_2.py that can be found at: https://github.com/TheFraserLab/WASP/tree/atac-seq-analysis/mapping. Briefly, for each read overlapping a SNP, we remapped hypothetical reads with the other allele, and discarded any reads that do not map uniquely, to the same location, for both alleles. Duplicate reads were filtered out for each replicate using https://github.com/eilon-s/bioinfo_scripts/blob/master/rmdup.py.

### Mapping and analyzing caQTLs

Pre- and post-ATAC allele frequencies, and the resulting p-values, were calculated using our published pipeline (*Tehranchi et al., 2016*). This uses post-ATAC allele frequencies together with individual sample genotypes to infer pre-ATAC allele frequencies.

To estimate pre-ATAC allele frequencies, the pre-ATAC pool could be sequenced; however this suffers from two major drawbacks. First, although sequencing the pre-ATAC pool could yield accurate pre-ATAC allele frequencies, it cannot account for genome-wide differences between samples such as the total amount of open chromatin. If one sample has more open chromatin than another, its alleles will be over-represented in the post-ATAC fraction. This will constitute a source of noise in our analysis, since our goal is to map *cis*-acting variation affecting individual sites. Second, sequencing the pre-ATAC pool would require very deep sequencing to achieve accurate allele frequency

estimates at each potential caQTL SNP, since the sequencing would not be restricted to SNPs in open chromatin regions (as they are for the ATAC fraction).

Therefore, we previously developed an alternative which does not require any additional sequencing, and does account for genome-wide differences between samples. Our regression-based approach (*Tehranchi et al., 2016*) uses the post-ATAC frequencies together with genotypes of each sample to infer the proportion of each sample in the pool. These proportions will be weighted by any genome-wide differences, since these will be naturally incorporated into the post-ATAC frequencies used as input to the regression. In this way, our pre-ATAC allele frequencies already account for some types of *trans*-acting variation, increasing our power for mapping *cis*-acting differences.

In the regression, our goal is to estimate the pooling weights (i.e. fraction of each individual in the pool, accounting for global *trans*-acting differences between samples). Using $p_i$ to denote the proportion of the minor allele of SNP $i$ in the pool:

$$p_i = \sum_{j=1}^{n} w_j G_{ij}$$

where $w_j$ is the pooling weight of the j'th individual, and $G_{ij}$ is the genotype of the j'th individual at the i'th SNP, coded as minor allele dosages (0, 0.5 or 1).

## Estimation of weights

Assuming that most SNPs are not caQTLs, our post-ATAC frequencies can be considered as unbiased estimates $p_i$ of $p_i$ for many SNPs (with the exception of any bias due to *trans*-acting variation that we seek to eliminate, as explained above). Our goal is to estimate the weights. From the equation above, one strategy to estimate the weights is to use linear regression, and regress the estimates onto the genotypes to obtain estimates of the weights $w_1, \ldots, w_n$. We used weighted linear regression with weights proportional to the inverse square-root of the read depth, since allele frequencies estimated from SNPs with more reads are likely to have less sampling error (however the results are similar if non-weighted regression or logistic regression is used instead).

Note that this approach ignores the set of constraints on the pooling weights, namely: $0 \leq w_j \leq 1$ for all $j$ and $\sum w_j = 1$. The former constraint is effectively not an issue, since in practice all of the estimates are in the legitimate range. To handle the second constraint, we replace $w_n$ by $1 - \sum_{j=1}^{n-1} w_j$. Rearranging we get:

$$E[\hat{p}_i - G_{in}] = \sum_{j=1}^{n-1} w_j (G_{ij} - G_{in})$$

So we can still use linear regression to estimate $w_1, \ldots, w_{n-1}$. This is done by regressing $\hat{p}_i - G_{in}$ onto $G_{i1} - G_{in}, \ldots, G_{i,n-1} - G_{in}$. The estimate of the missing weight is given by $\hat{w}_n = 1 - \sum_{j=1}^{n-1} \hat{w}_j$.

## Inference of pre-ATAC allele frequencies

Given that the estimated weights are not too close to 0 or 1, the joint distribution of the estimates of the weights follows approximately:

$\hat{w}_{-n} \sim MVN(w_{-n}, \Sigma)$,

where $\Sigma$ is the covariance matrix (which is given by the regression) and $w_{-n}$ is the vector of weights without the n'th coordinate. We estimate the pre-ATAC proportions as

$$\hat{p}_i^{pre} = \sum_{j=1}^{n} \hat{w}_j G_{ij},$$

and the variance of the estimate is given by

$$V(\hat{p}_i^{pre}) = V\left(\sum_{j=1}^{n} \hat{w}_j G_{ij}\right) = V\left(\sum_{j=1}^{n-1} \hat{w}_j G_{ij} + \left(1 - \sum_{j=1}^{n-1} \hat{w}_j\right) G_{in}\right) =$$

$$\sum_{j,k=1}^{n-1} G_{ij}G_{ik}\Sigma_{jk} + \sum_{j,k=1}^{n-1} G_{in}^2 \Sigma_{jk} - 2\sum_{j,k=1}^{n-1} G_{ij}G_{in}\Sigma_{jk}.$$

As a sanity check, note that in the hypothetical case when all the genotypes are identical, the variance of the pre-ATAC proportion is 0, and indeed when all the genotypes are the same, the pre-ATAC proportion would be the same for any set of weights, so no uncertainty is introduced by not knowing the weights.

## Testing for difference between pre-ATAC and post-ATAC proportions

Having estimated the pre-ATAC and post-ATAC allele frequencies, our next goal is to identify significant differences while accounting for uncertainty in the pre-ATAC estimates. The binomial distribution is a natural choice for this, if its assumptions are met. To test whether the binomial distribution is appropriate, we fit a negative binomial regression and observed that the estimated dispersion parameter was very close to 1, hence the depth (after duplicate removal) follows a Poisson distribution, and the binomial distribution is appropriate (*Golan and Rosset, 2013*).

Conditional on the number of reads covering each SNP $n_i$, we model the number of minor allele reads $m_i$ as following a binomial distribution $Bin(p_i, n_i)$. Under the null hypothesis of no change between the pre-ATAC and post-ATAC pools we have $p_i = p_i^{pre}$.

One possibility is to standardize the pre- and post-ATAC proportion estimates and to apply a Z-test. However, we found that this test results in an inflated rate of type-1 errors, since the normal approximation is not always adequate for the post-ATAC reads, especially when $n_i$ or $p_i^{pre}$ are small. Instead, we compute the p-value directly using numerical integration. First, imagine that $p_i^{pre}$ were known. In that case, the p-value is given by:

$$pv(p_i^{pre}) = \sum_{j=0}^{n_i} \binom{n_i}{j} (p_i^{pre})^j (1 - p_i^{pre})^{n_i - j} l\left\{ \left| \frac{m_i}{n_i} - \frac{j}{n_i} \right| \geq \left| p_i^{pre} - \frac{m_i}{n_i} \right| \right\}.$$

In other words, we sum the binomial probabilities for any possible number of reads that is as far or further from the expected proportion under the null hypothesis. To make this more concrete, consider the following toy example: the pre-ATAC proportion is 0.5, the read depth is 10, and we observe 3 reads with the minor allele. The p-value is defined as the probability of observing a value that is as likely or less likely than the actual observed value, under the null hypothesis. Here we expect to see 5 reads with the minor allele. Since the alternative hypothesis is two-sided (we do not know which allele will increase post-ATAC), being less likely means being further away from 5. Hence, the p-value is calculated as the probability of observing 0, 1, 2, 3, 7, 8, 9 or 10 reads with the minor allele, under the null hypothesis that the number of minor allele reads follows a $Bin(n = 10, p = 0.5)$ distribution. However, $p_i^{pre}$ is not known. One option is to plug in the estimate $p_i^{pre}$, but this practice ignores the variance of the estimate, and will result in optimistic p-values. Instead, we integrate over all possible values of $pv(p_i^{pre})$ to get the actual p-value, thus factoring in the variance of the estimate:

$$pv = \frac{1}{C} \int_0^1 \frac{1}{\sqrt{2\pi v_i}} e^{-\frac{1}{2v_i}\left(p - \hat{p}_i^{pre}\right)} pv(p) dp,$$

where:

$$C = \int_0^1 \frac{1}{\sqrt{2\pi v_i}} e^{-\frac{1}{2v_i}\left(p - \hat{p}_i^{pre}\right)} dp$$

is a normalizing constant, meant to deal with cases where the tails of the normal distribution that fall outside of the [0,1] are non-negligible (which could result in optimistic p-values), and $v_i = V(\hat{p}_i^{pre})$.

SNPs were considered testable in a population if they were covered by at least 20 ATAC-seq reads and had minor allele frequency >0.01 in that population. No peak calling was performed; each SNP was tested for allelic bias using only the reads overlapping it, since these reads are the only ones that give information about its allelic bias (*Tehranchi et al., 2016*). Genotypes were downloaded from the 1000 Genomes Project. Scripts and documentation can be found at: https://github.com/tehranchi/public.

Directionality and enrichment tests were performed as described in *Tehranchi et al., 2016*. Numbers underlying specific analyses are reported in source data files provided for all figures.

## Fisher's combined probability test

Shared caQTLs were calculated using Fisher's combined probability test, where $p_i$ is the caQTL p-value for population $i$. Any SNP not tested in population $i$ was assigned $p_i = 1$.

$$X_{2k}^2 = -2 \sum_{i=1}^{k} \ln(p_i)$$

## Motif analysis

We used HOMER (*Heinz et al., 2010*) to search for motifs that were differentially enriched among the high-CA alleles compared to the low-CA alleles in our shared caQTLs. The caQTL plus 15 bp on each side (31 bp total) was used as input to HOMER, with the less accessible alleles of the same caQTLs used as the background comparison set. Therefore all significant enrichments are due to caQTL variants within the motifs themselves (since any motifs flanking the caQTLs would be present in both comparison sets). We ran the findMotifs.pl script from the HOMER package where the targetSequences.fa file contained the high-CA alleles and the background.fa file contained the low-CA alleles, and then repeated the analysis with the two files swapped: findMotifs.pl <targetSequences. fa> fasta < output directory> -fasta <background.fa>

## Long-range interaction analysis

To obtain *Figure 4A–B*, we restricted the analysis to shared caQTLs (Fisher's combined p<5×10$^{-6}$) with consistent directionality in a majority of the populations tested, and counted the number of Hi-C reads connecting each caQTL allele with any distal locus (>20 kb or >100 kb away). In order to use only allele-specific Hi-C reads, we restricted the analysis to read pairs where at least one read overlaps a heterozygous variant in the GM12878 cell line (as described in *Rao et al., 2014*). We then determined the number of caQTLs where the more accessible allele had significantly more Hi-C contacts (using a binomial test of allele-specific read counts), compared to the number where the less accessible allele was more interactive. See *Figure 4—source data 1* for detailed results.

For *Figure 4C* we first obtained two subsets of eQTLs (*Lappalainen et al., 2013*): those that are proximal to their target TSSs (<10 kb) and those that are distal (>100 kb). Next, we calculated the enrichments of these proximal and distal subsets among shared caQTLs. To obtain the p-values in *Figure 4C*, we used Fisher's exact test to compare the significance of these enrichments with the respective enrichments of proximal and distal eQTLs among non-caQTLs (Fisher's combined p>0.5). We repeated this analysis on the subsets of caQTLs and non-caQTLs in regions without significant allelic bias (binomial p>0.5) in the number of reads supporting long range (>100 kb) interactions, requiring each SNP to have at least 100 reads supporting long-range interactions in order to be confident in the lack of strong allelic bias. The non-significant (p=0.20) enrichment in *Figure 4C* is based on a relatively large number of SNPs (1680 caQTLs with no effect on interactions, and 5719 non-caQTLs with no effect on interactions)—the same background set as used for the significant (p<10$^{-70}$) enrichment in the same figure—suggesting that a small sample size is not driving the lack of significant enrichment. See *Figure 4—source data 1* for further details of this analysis.

## GWAS overlaps

We used all reported GWAS associations at p<5×10$^{-8}$ from the GRASP database (28), excluding those for gene expression levels (eQTLs). In order to include cases where the reported SNP is not the causal variant, we expanded the GWAS list to include any variants in strong LD with GWAS hits ($r^2$ >0.8 in the CEU population; CEU is similar to the European cohorts in which nearly all of the GWAS were conducted). We then intersected this LD-expanded list with our caQTLs (*Supplementary file 3*). Because most GWAS hits are in strong LD with other variants, many of the caQTLs that overlap GWAS hits may not be causal SNPs for the GWAS trait; we cannot apply published co-localization methods to quantify this, since these require association data across an entire locus or LD block for comparison, and our pooled approach is limited to SNPs covered by the ATAC-seq reads.

## IDR analysis

For IDR calculation, we used parameters similar to the ENCODE project (mu = 0.1, sigma = 1.0, rho = 0.2, p=0.5, eps = $10^{-6}$, max.ite = 3000). Since IDR does not allow tied p-values, we broke ties by adding a small random number (normally distributed with zero mean and standard deviation $10^{-17}$).

## Controlling for ATAC-seq read density

In Figure 3A we present the chromatin states of caQTLs compared to the genome as a whole. Much of the difference is driven by ATAC-seq read density; that is we would see strong enrichments for TSS and enhancers in any ATAC-seq experiment. To ask the more specific question of where caQTLs are enriched or depleted after controlling for read density, we selected non-caQTL variants matched for the number of reads to each shared caQTL, and then quantified the difference in chromatin state enrichments for shared caQTLs vs read-matched non-caQTLs. Details of read-matching are given in the 'Effects of caQTLs on DNA shape' section below.

## Effects of caQTLs on DNA shape

Since the shape of the DNA double helix is sequence-dependent, and can affect interactions with TFs (*Chiu et al., 2016*), we tested whether shared caQTL variants tend to cause larger changes in DNA shape than non-caQTLs. To compare caQTLs vs non-caQTLs, we defined a non-caQTL as a variant with caQTL p=1 and at least 20 ATAC-seq reads. We first computed the total read depth for each shared caQTL and non-caQTL across all populations. We then created a read-matched set of shared caQTLs and non-caQTLs in the following way: For each shared caQTL, we identified a unique non-caQTL where | log$_2$(shared caQTL reads) - log$_2$(non-caQTL reads) |<0.5. If no such non-caQTL existed, we eliminated the shared caQTL. We were able to find unique read-matched non-caQTLs for 11,416 of the shared caQTLs.

After identifying read-matched shared caQTLs and non-caQTLs, we used BEDTools version 2.26.0 (*Quinlan and Hall, 2010*) to extract the genome sequence at each variant ±5 bp. For each shared caQTL, we created four sequences – one with the open allele ±5 bp, one with the closed allele ±5 bp, one with the read-matched non-caQTL reference allele ±5 bp, and one with the read-matched non-caQTL alternate allele ±5 bp. We then used DNAshapeR version 1.0.2 (*Chiu et al., 2016*) to estimate the minor groove width (MGW), propeller twist (ProT), Roll, and helix twist (HelT) for each of the sequences for each shared caQTL. We computed the difference in each shape parameter between the sequences for the open and closed caQTL alleles as well as between alleles for the read-matched non-caQTLs, where the 'open' and 'closed' alleles were randomly selected. To compute whether the shared caQTLs have a stronger association with DNA shape than the read-matched non-caQTLs, we used the Wilcoxon rank-sum test to compare the two difference distributions at the position of the variant. Since Roll and HelT shapes are computed in groups of two nucleotides, we concatenated the difference distributions for the two windows of 2 bp that overlap the variant. We multiplied each p-value by four as a Bonferroni correction.

## Supplementary Text

### Assessing caQTL reproducibility with IDR

A standard approach for estimating an FDR in association studies is to randomize the data, for example randomly pairing one sample's genotypes with another sample's ATAC-seq data, and then recalculate the associations to gauge the extent of false positives. Since our approach does not have separate data for each individual, such randomization is not possible, and thus we cannot estimate an FDR for our caQTLs using standard approaches. However, we can apply a related method known as the irreproducible discovery rate (IDR), developed as part of the ENCODE project (*Li et al., 2011*; *ENCODE Project Consortium, 2012*) to assess the statistical reproducibility of our caQTL p-values. To measure this, we calculated caQTL p-values separately for each biological replicate of two populations, ASW and CEU. The IDR estimates the fraction of data points that are not reproducible at any given p-value cutoff. For a range of potential cutoffs, we observed a similar trend in both populations: at p=5×$10^{-3}$ we observed IDR ≈ 0.03, and p=5×$10^{-4}$ corresponds to IDR ≈ 0.01 (dashed red line in *Figure 1—figure supplement 3*). These numbers are consistent with

other results suggesting a low FDR among our caQTLs, such as their ~ 97–99% agreement with dsQTL and bQTL directionality (*Figure 1E–F*).

## Comparison to non-pooled chromatin accessibility studies

In our comparison with RASQUAL caQTLs (*Kumasaka et al., 2016*), we first sought to identify a matched significance cutoff. Among their 2707 caQTLs (FDR = 10%), 3.5% were previously identified as dsQTLs (*Degner et al., 2012*) (the most closely matched type of QTL available for comparison). This 3.5% overlap between two previous studies of chromatin accessibility QTLs may seem low, but likely results from differences in experimental methods (ATAC-seq vs DNase-seq), analysis methods, and populations, as well as incomplete power. In CEU, our population most closely matched to the British (GBR) population used for RASQUAL, we found that our top 7587 caQTLs ($p<3\times10^{-6}$) yielded the same overlap of 3.5% dsQTLs. Our approach does not involve calling CA peaks, but in a separate analysis we estimated that <20% of our caQTLs fall into the same peak as another caQTL, so removing these would not have a major effect on the results.

In addition to the RASQUAL comparison, we also compared our caQTLs to variants associated with allelic imbalance of chromatin accessibility (*Maurano et al., 2015*). In that study, 493 samples from 166 individuals were profiled with a total of $2.6 \times 10^{10}$ DNase-seq reads. At the most stringent cutoff (FDR = 0), they called 2420 imbalanced SNPs, 2.1% of which were also dsQTLs. Comparing this to our study, our top 16,990 CEU caQTLs ($p<3.5\times10^{-4}$) had the same 2.1% enrichment. Therefore we mapped 7-fold more caQTLs from 246-fold fewer libraries and 443-fold fewer reads, though their inclusion of diverse tissues likely decreased the overlap with dsQTLs from LCLs.

We note that dsQTLs are not the only possible benchmark we could use to select caQTL lists of equal quality across studies. When using CTCF bQTLs (*Ding et al., 2014*) instead, our estimated improvement in cost per QTL was even greater than when using dsQTLs, due to the high enrichment of CTCF bQTLs in our data (~190 fold in our CEU caQTLs [*Figure 1—figure supplement 5*], compared to ~7 fold in RASQUAL caQTLs (*Kumasaka et al., 2016*) and 4.5-fold in the imbalanced DNase-seq SNPs (*Maurano et al., 2015*).

## Comparing selection against caQTLs and nonsynonymous variants

If this depletion of caQTLs near TSSs (*Figure 3A*) is due to selection against deleterious effects of these caQTLs, then we can compare this with human Ka/Ks ratios, which reflect the depletion of nonsynonymous changes in protein-coding regions. Across all human genes, Ka/Ks since divergence with chimpanzee was estimated as 0.208 (*Chimpanzee Sequencing and Analysis Consortium, 2005*), implying ~4.8 fold depletion for nonsynonymous changes. Therefore we observed 8.7/ 4.8 = 1.81 fold (i.e. 81%) greater depletion for the caQTLs. This comparison is only an approximation, since it ignores effects of positive selection, as well as the possibility that caQTLs are depleted near TSSs for reasons other than selection (e.g. CA having greater robustness to mutations near TSSs). We also note that if there is any negative selection against TSS-proximal SNPs that are not caQTLs then the 1.81-fold difference will be an underestimate.

## Causality of joint bQTLs/caQTLs affecting long-range chromosomal interactions

In addition to CTCF, we previously reported an association between long-range chromosomal interactions and binding of five additional TFs in LCLs (*Tehranchi et al., 2016*). However, in that work we did not explore the mechanism by which this occurs or the direction of causality: it could be that TF binding impacts long-range interactions, though other causal scenarios are also possible. For example, joint bQTLs/caQTLs could be cases where a variant first affects TF binding and then CA as a result; or the variant's direct effect could be on CA, which then changes the local landscape of TF binding; or there may be independent effects on TF binding and CA that are not causally linked (either via two distinct causal variants in linkage disequilibrium, or one variant with independent effects on TF binding and CA). In general, the strong enrichment for overlap between caQTLs and bQTLs suggests that causal independence is rare at these joint QTLs, since under such a model one would not expect to see either strong enrichment for overlap, or the nearly perfect concordance in directionality that we observed (~98–99%). In most cases we cannot distinguish between the first two scenarios, that is whether the TF binding or CA effect 'comes first'; however in cases where a

joint bQTL/caQTL disrupts a known TF binding motif, then we can infer that its primary effect is likely to be on TF binding to DNA, and thus any other effects—on CA or other traits—are likely to be downstream consequences of this.

We used this logic to ask whether TF binding affects long-range chromosomal interactions, by restricting our test for long-range interactions only to those caQTLs that disrupt a known TF binding motif and are present within a DNase hypersensitive footprint indicative of TF binding (*Degner et al., 2012*; *Maurano et al., 2015*); these caQTLs are most likely caused by the SNP's direct effect on TF binding having a subsequent effect on CA. For these caQTLs, we observed a statistically indistinguishable enrichment for the more accessible allele having more interactions (63/24 = 2.6 fold), indicating that disruptions of TF binding lead to changes in both CA and long-range chromosomal interactions. We therefore show TF binding upstream of CA in *Figure 4D*, though the reverse causality could also occur in some cases.

We could not test the possibility that some other variants may have a primary effect on CA and secondary effects on TF binding, since we cannot determine which caQTLs are not bQTLs for any TF, given the limited amount of bQTL information available (*Figure 4—figure supplement 2*).

## Regulatory effects of caQTLs across tissues

To explore the effects of caQTLs on gene expression across tissues, we intersected our caQTLs with eQTLs mapped in a meta-analysis across 44 tissues/cell types (*Battle et al., 2017*). Each eQTL was assigned a posterior probability of being active in each tissue, and following the GTEx authors, we required p>0.9 (i.e. 90% probability) to call an eQTL as likely active in a given tissue. Overall, the median number of tissues per eQTL was 19 out of 44. In contrast, among 2833 caQTLs that were also eQTLs, the median number of active tissues was 25/44 (Wilcoxon p=$4\times10^{-21}$). This implies that the underlying caQTL effects are probably shared across most tissues as well, with 25/44 being a likely underestimate of the extent of tissue sharing, since caQTLs present in a tissue do not necessarily act as eQTLs (as we have seen for LCLs).

We then asked in what tissues/cell types were our caQTLs most likely to affect gene expression. To measure this, we calculated the ratio of [fraction of joint eQTL and caQTL with posterior p>0.9 in tissue X] / [fraction of all eQTLs with posterior p>0.9 in tissue X]; values greater than one indicate that the joint eQTL and caQTLs are more likely to affect expression in tissue X than are eQTLs overall (this accounts for differences in numbers of eQTLs per tissue, which is largely driven by sample sizes). All 44 tissues had values > 1.06, with two clear outliers, LCLs and whole blood, having the highest chance of caQTLs affecting expression (*Figure 4—figure supplement 4*); this is not surprising because LCLs were used in our study and whole blood is the most similar tissue to LCLs in this dataset. At the other extreme, the ten sampled brain regions and testes had the 11 lowest ratios, indicating that caQTLs active in LCLs are slightly less likely to affect gene expression in these tissues.

## caQTLs in one cell type can underlie eQTLs specific to other cell types/conditions

Our multi-tissue analysis above (*Figure 4—figure supplement 4*) suggested that caQTLs from LCLs can be informative about transcriptional regulation in other cell types. We hypothesized that even in cases where a caQTL is not an eQTL in LCLs, if the effect on CA is preserved across other cell types or conditions, then a change in the expression of a trans-acting factor could cause a constitutive caQTL to be a condition-specific eQTL. Indeed, in our multi-tissue analysis, we observed 200 caQTLs that had a posterior p<0.1 of being an eQTL in LCLs, but nevertheless were called as likely eQTLs (p>0.9) in an average of 11.4 other tissues. We have also come across several examples of this outside of the GTEx data, and highlight two of these below.

A previous study identified a SNP, rs9806699, that acts as an eQTL for *GATM* in LCLs only after exposure to the statin drug simvastatin (*Mangravite et al., 2013*). This SNP was also associated with statin-induced myopathy in two cohorts, though the molecular mechanism underlying this condition-specific eQTL was not investigated. We found that this same SNP is also a caQTL in the GWD population (p=$6.3\times10^{-5}$), which may underlie its statin-dependent transcriptional response.

Another example is a caQTL that is one of five linked variants associated with rheumatoid arthritis, all in strong LD ($r^2 >0.95$) and with similar GWAS p-values ($1.3 \times 10^{-7} < p<3.1\times10^{-7}$) that preclude fine-mapping (*Farh et al., 2015*). The caQTL is in a 'super-enhancer' (long enhancers bound

by many TFs) 6 kb upstream of *LBH*, a regulator of synoviocyte proliferation (*Ekwall et al., 2015*). This same variant was recently shown to causally affect *LBH* transcription in fibroblast-like synoviocytes (*Hammaker et al., 2016*), with the less accessible allele associated with reduced expression (as well as higher RA risk). This variant has not been found as an eQTL for *LBH* in LCLs (*Battle et al., 2017*; *Lappalainen et al., 2013*).

These two examples illustrate how caQTLs mapped in one cell type/condition can have transcriptional effects that are specific to other cell types/conditions.

## Acknowledgements

We would like to thank members of the Fraser lab, J Pritchard, and W Greenleaf for helpful advice, and M Simon for sharing lab space. This work was supported by NIH grant 2R01GM097171-05A1.

# Additional information

### Funding

| Funder | Grant reference number | Author |
|---|---|---|
| National Institute of General Medical Sciences | 2R01GM097171-05A1 | Hunter B Fraser |

The funders had no role in study design, data collection and interpretation, or the decision to submit the work for publication.

### Author contributions

Ashley Tehranchi, Formal analysis, Writing—review and editing, Experiments; Brian Hie, Michael Dacre, Irene Kaplow, Kade Pettie, Peter Combs, Formal analysis, Writing—review and editing; Hunter B Fraser, Conceptualization, Formal analysis, Supervision, Funding acquisition, Writing—original draft, Writing—review and editing

### Author ORCIDs

Michael Dacre (ID) http://orcid.org/0000-0002-5561-1656
Peter Combs (ID) https://orcid.org/0000-0003-2835-5623
Hunter B Fraser (ID) https://orcid.org/0000-0001-8400-8541

### Decision letter and Author response

Decision letter https://doi.org/10.7554/eLife.39595.032
Author response https://doi.org/10.7554/eLife.39595.033

# Additional files

### Supplementary files

• Supplementary file 1. All cell lines used, and all caQTLs at a nominal $p<5\times10^{-4}$. The numbers of caQTLs are greater than those shown in *Figure 1B*, since for *Figure 1B* and the associated text we removed those in LD ($r^2 >0.8$ in YRI).
DOI: https://doi.org/10.7554/eLife.39595.023

• Supplementary file 2. Shared caQTLs at a Fisher's combined $p<5\times10^{-6}$.
DOI: https://doi.org/10.7554/eLife.39595.024

• Supplementary file 3. All caQTL/GWAS overlaps. Rows where the GWAS rsID is the same as the caQTL rsID indicate that the GWAS variant was itself a caQTL; rows where they do not match indicate the two variants are in LD ($r^2 >0.8$ in CEU). caQTLs with $p<0.005$ were included.
DOI: https://doi.org/10.7554/eLife.39595.025

• Transparent reporting form
DOI: https://doi.org/10.7554/eLife.39595.026

## Data availability

All ATAC-seq reads are available at NCBI SRA, project ID PRJNA383900. All genome sequence data are available at http://www.internationalgenome.org.

The following dataset was generated:

| Author(s) | Year | Dataset title | Dataset URL | Database and Identifier |
|---|---|---|---|---|
| Ashley Tehranchi | 2018 | ATAC-seq in 10 populations | https://www.ncbi.nlm.nih.gov/sra/?term=PRJNA383900 | Sequence Read Archive, PRJNA383900 |

The following previously published datasets were used:

| Author(s) | Year | Dataset title | Dataset URL | Database and Identifier |
|---|---|---|---|---|
| 1000 Genomes Project | 2015 | Genome sequence data from the 1000 Genomes Project | ftp://ftp.1000genomes.ebi.ac.uk/vol1/ftp/release/20130502/ | 1000 Genomes Project, 20130502 |

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
