## [Decision Letter]

[**Editorial note:** This article has been through an editorial process in which the authors decide how to respond to the issues raised during peer review. The Reviewing Editor's assessment is that minor issues remain unresolved.]

Acceptance letter:

The Reviewing Editor's assessment is that a minor issue (the appropriateness of the term "fine-mapping") remains unresolved.

Decision letter after peer review:

Thank you for submitting your article "Fine-mapping *cis*-regulatory variants in diverse human populations" for consideration by *eLife*. Your article has been reviewed by two peer reviewers, and the evaluation has been overseen by a Reviewing Editor and Patricia Wittkopp as the Senior Editor. The reviewers have opted to remain anonymous.

The Reviewing Editor has highlighted the concerns that require revision and/or responses, and we have included the separate reviews below for your consideration. If you have any questions, please do not hesitate to contact us.

Summary:

The authors describe a novel approach for the identification of chromatin accessibility QTLs (caQTLs) on the basis of ATAC-seq data obtained from pooled sequencing in LCLs from thousands of individuals across 10 population groups. The results of these analyses are used to assess allelic effects on chromatin across populations, and demonstrate that chromatin accessibility is associated with transcription factor binding, long-range chromatin interactions and gene expression. The authors then propose the use of variants with shared allelic effects on chromatin across populations as an approach to prioritise causal variants for complex traits at loci identified through GWAS. Both reviewers highlighted the novelty and importance of the study in understanding how genetic variation impacts on complex traits via chromatin accessibility across population groups.

Major concerns:

The reviewers expressed concerns over the interpretation of the results of the fine-mapping analyses. To comprehensively fine-map a GWAS locus, the effect of all (or most) variants on a trait are assessed, but the approach can only consider variants with sufficient ATAC-seq read depth. It is also assumed that causal variants will have shared allelic effects on chromatin across populations. This does not allow for potential allele frequency differences between populations which might impact on power to detect allelic imbalance in each population. Whilst the approach can be used to prioritise potential causal variants, it is not "fine-mapping" in the traditional sense, and the authors should consider these limitations in the interpretation of their results.

The reviewers also felt that some further discussion of the potential limitations of the approach, and the impact on the interpretation of their findings, would be warranted, including the ability to only detect caQTLs that affect their own read counts, the reliability of estimation of pre-ATAC/ChIP allele frequencies and potential mapping bias of Hi-C reads.

Separate reviews (please respond to each point):

*Reviewer #1:*

The manuscript by Tehranchi et al. describes the generation of ATAC-seq data in LCLs from a thousand individuals across 10 populations using a pooled approach and then mapping allelic effects in pooled data from each population. The authors then use these results to describe relationships in allelic effects on chromatin across populations, identify variants with shared effects across populations, compare allelic effects to other molecular trait data, and use these variants to prioritize fine-mapped GWAS variants. In total this study and findings are both an important resource and provide a conceptual advance in our understanding of how population genetic variation affects chromatin accessibility and role in complex traits and disease.

The authors describe their approach of prioritizing variants with significant allelic effects across populations as fine-mapping causal variants, which I think is not entirely accurate. The premise of fine-mapping is that most/all variants on in a region are evaluated for their effects on a trait; given that the authors are only able to map the subset of variants with sufficient ATAC-seq read depth there may be causal variants that, for example, map in a different site and affect chromatin accessibility across an entire region, map in the same site but on the edge and have low read depth, or map outside of a site entirely. While presumably the majority of variants with shared allelic effects are likely directly responsible for their imbalance, there are likely also variants where this isn't the case.

Furthermore, assuming that only variants with shared effects across populations are causal for a trait also does not consider that (a) some trait/disease signals have population-specific effects and (b) allele frequency differences across populations might prohibit mapping allelic imbalance in all populations and lead to the assumption that the effects are population-specific. In terms of mapping causal variants at disease signals then, this approach would work well if the signal is shared across populations with consistent frequency, but if it is population-specific or has allelic heterogeneity then this approach might not be as applicable.

It would help the study for the authors to be more explicit about these limitations, and re-frame the description of the results to clarify that they aren't fine-mapping causal variants in a classic sense but rather identifying a specific set of variants with shared effects across populations that are extremely useful in interpreting fine-mapped disease signals which are shared across populations.

For the GWAS examples it would be extremely useful to readers I think to see plots of the regions including the data described for each example, which I couldn't find (maybe I missed them). For the second example, it would be useful to further see the LD patterns across populations to demonstrate that rs479844 is associated with Europeans but not in other populations presumably due to differences in LD with rs10791824.

Minor Comments:

1) When comparing the percentage of variants with allelic effects shared across populations, presumably they considered a variant shared across two populations if it had significant effects in both? If so it would be informative to estimate how many variants were also likely shared but didn't reach significance for example by estimating concordance in effects.

2) It would be interesting to determine whether variants with shared allelic effects tended to have higher causal probabilities from genetic fine-mapping than other classes of variants to provide further support for their causality.

3) The statement "An alternative to increasing sample size within a single population is trans-ethnic fine-mapping, in which a GWAS is performed across multiple populations" is confusing and could be clarified.

4) I think purists might quibble with the use of QTL to describe allelic imbalance mapping, and could also potentially cause confusion to readers not overly familiar with this field.

*Reviewer #2:*

In Tehranchi et al., 2018, the authors present a novel use of pooled sequencing applied to the identification of QTLs altering chromatin accesibility (caQTLs) by ATAC-seq. They demonstrate convincingly that their pooled approach can identify QTLs more efficiently than standard approaches, that CA is causally associated with transcription factor binding, long-distance chromatin interactions, and gene expression, and that chromatin accessibility is a likely molecular trait underlying GWAS variants. However, there are several concerns:

1) caQTLs can only be detected if they affect their own read counts. Is this a major limitation of the method given the findings of previous studies? This should be addressed in the text.

2) Do caQTLs show different allele frequencies across genomic annotations (e.g. active TSS vs enhancers vs repressed regions, etc)? Does this contradict or support hypotheses of negative selection on caQTLs?

3) Have the authors validated, in this or previous work, that they are able to reliably estimate the pre-ATAC/ChIP allele frequencies using their regression approach?

4) For the "Long-range interaction analysis," is potential mapping-bias of Hi-C reads taken into account by the authors?

Minor Comments:

1) Introduction section: while most GWAS variants are not in LD with coding variants or may be in LD with variants in regulatory regions, most genome-wide significant variants from GWAS aren't actually in regulatory regions.

2) In "Characterizing fine-mapped eQTLs," the source data of the chromatin states (Ernst and Kellis, 2012) is present in the figure legend but is absent in the text and methods; adding a citation to one of these would be helpful

3) As a minor stylistic concern, chromHMM tracks have a standard color palette (e.g. red corresponds to promoter/TSS, orange/yellow to enhancer, green to transcribed regions, etc.) and this reviewer would prefer its use for Figure 3.

4) It is not clear how read density was controlled for in subsection “Characterizing fine-mapped caQTLs”.

5) The final paragraph of subsection “Characterizing fine-mapped caQTLs” references "trinucleotide" dependencies of CA variants but these are not mentioned in the text or Materials and methods section.

6) The figure legend for Figure 4 mentions using "allele-specific 3D chromosomal interaction" data; does this mean only heterozygous sites were considered? This should be described in more detail in the text and/or methods.

7) Having the principal equations of the regression approach in the Materials and methods section would be helpful to readers.

8) The subsection "Effects of caQTLs on DNA shape" in the Supplementary Text includes analysis related to chromatin shape, which is not present in the main text.

Additional data files and statistical comments:

The provided data files, in addition to read data placed on public databases, should be sufficient.

[Editors' note: further revisions were requested prior to acceptance, as described below.]

Thank you for submitting a revised version of your article "Fine-mapping *cis*-regulatory variants in diverse human populations" for consideration by *eLife*. Your article has been reviewed by the original referees and the evaluation has been overseen by Andrew Morris as the Reviewing Editor and Patricia Wittkopp as the Senior Editor.

The manuscript has been improved, but the reviewers remain concerned about the use of the term "fine-mapping" in the manuscript. Traditional fine-mapping, in the GWAS field for example, interrogates all (or the majority) of variants in a region, and assesses their relative evidence for causality based on association with a trait and relevant annotation. However, in your investigation, only a limited subset of variants in a region are considered, and the reviewers do not feel that "fine-mapping" is an appropriate description of the approach taken. The reviewers have suggested the following changes be made.

1) Remove the term "fine-mapping" from the title of the manuscript – this could be replaced by "determining" or "localising".

2) Please provide further discussion of the limitation of the approach to localising causal variants. In particular, please include discussion to address the following comment from the initial review of your manuscript:

"The authors describe their approach of prioritizing variants with significant allelic effects across populations as fine-mapping causal variants, which I think is not entirely accurate. The premise of fine-mapping is that most/all variants on in a region are evaluated for their effects on a trait; given that the authors are only able to map the subset of variants with sufficient ATAC-seq read depth there may be causal variants that, for example, map in a different site and affect chromatin accessibility across an entire region, map in the same site but on the edge and have low read depth, or map outside of a site entirely. While presumably the majority of variants with shared allelic effects are likely directly responsible for their imbalance, there are likely also variants where this isn't the case."

---

## [Author Response]

Reviewer #1:

[…] The authors describe their approach of prioritizing variants with significant allelic effects across populations as fine-mapping causal variants, which I think is not entirely accurate. The premise of fine-mapping is that most/all variants on in a region are evaluated for their effects on a trait; given that the authors are only able to map the subset of variants with sufficient ATAC-seq read depth there may be causal variants that, for example, map in a different site and affect chromatin accessibility across an entire region, map in the same site but on the edge and have low read depth, or map outside of a site entirely. While presumably the majority of variants with shared allelic effects are likely directly responsible for their imbalance, there are likely also variants where this isn't the case.Furthermore, assuming that only variants with shared effects across populations are causal for a trait also does not consider that (a) some trait/disease signals have population-specific effects and (b) allele frequency differences across populations might prohibit mapping allelic imbalance in all populations and lead to the assumption that the effects are population-specific. In terms of mapping causal variants at disease signals then, this approach would work well if the signal is shared across populations with consistent frequency, but if it is population-specific or has allelic heterogeneity then this approach might not be as applicable.It would help the study for the authors to be more explicit about these limitations, and re-frame the description of the results to clarify that they aren't fine-mapping causal variants in a classic sense but rather identifying a specific set of variants with shared effects across populations that are extremely useful in interpreting fine-mapped disease signals which are shared across populations.

We agree that our fine-mapping approach is distinct from traditionally used methods. We have added the following to clarify this in the Discussion: “our fine-mapping approach is distinct from typical GWAS fine-mapping where every variant in a locus is genotyped in a large cohort, and like all forms of trans-ethnic fine-mapping, our method will only work when a variant is present at an appreciable frequency, and affects the same trait, across multiple populations.”

For the GWAS examples it would be extremely useful to readers I think to see plots of the regions including the data described for each example, which I couldn't find (maybe I missed them). For the second example, it would be useful to further see the LD patterns across populations to demonstrate that rs479844 is associated with Europeans but not in other populations presumably due to differences in LD with rs10791824.

We have added the LD information to the Results: “A meta-analysis reported that when non-European ethnicities are included, the only consistent association is for another nearby SNP, rs10791824 (Paternoster et al., 2015), which is in strong LD with rs479844 in Europeans (*r*2 =0.92) but not in our four African populations (*r*2 = 0.21).” We have also added a new supplemental figure showing the causal probabilities of the SNPs at two of the loci discussed. We were not able to plot the third example, since only SNPs with >2.75% causal probability were reported, and 58 of the 62 SNPs in this LD block were below this cutoff.

Minor Comments:1) When comparing the percentage of variants with allelic effects shared across populations, presumably they considered a variant shared across two populations if it had significant effects in both? If so it would be informative to estimate how many variants were also likely shared but didn't reach significance for example by estimating concordance in effects.

Our method for combining signal across populations was based on Fisher’s combined p-values, which does not actually require significance in each individual population. Instead, the test estimates a single p-value representing the chance of observing all of the given p-values (in this case representing distinct populations) under the null of uniformly distributed p-values. For example, in Figure 2—figure supplement 2, we show a shared caQTL (rs79979970) that is individually significant in only one population (CHB) out of eight tested, but reaches a shared caQTL p =5.6x10-7 because it has p < 0.1 in an additional four populations. We have provided further details and an equation for this method in the Materials and methods section.

2) It would be interesting to determine whether variants with shared allelic effects tended to have higher causal probabilities from genetic fine-mapping than other classes of variants to provide further support for their causality.

We appreciate this suggestion, and have conducted this analysis. We added the following paragraph to our Results section: “In addition to revealing insights into transcriptional regulation, caQTLs also provide a means to explore genotype/phenotype associations by identifying likely causal variants and their molecular mechanisms of action. To compare our fine-mapping approach with standard GWAS fine-mapping, we asked whether our shared caQTLs were more likely to be assigned a high probability of being causal for disease risk (Farh et al., 2015). We found that for several autoimmune diseases—the class of disease most directly relevant to LCLs—shared caQTLs were highly enriched for likely causal variants (e.g. for Crohn’s disease, mean causal probability for shared caQTLs was 2.4-fold higher than for non-caQTLs, p = 0.002; for ankylosing spondylitis, mean causal probability for shared caQTLs was 2.6-fold higher, p = 0.001). However for most diseases, the number of overlaps between these two data sets was too small to conduct a meaningful test. Together with other evidence of the efficacy of our finemapping approach (Figure 2C), this suggests that our shared caQTLs are an effective means of resolving likely causal cis-regulatory variants.”

3) The statement "An alternative to increasing sample size within a single population is trans-ethnic fine-mapping, in which a GWAS is performed across multiple populations" is confusing and could be clarified.

We have modified this: “To achieve high mapping resolution, an alternative to increasing sample size within a single population is trans-ethnic fine-mapping, in which a GWAS is performed across multiple populations”

4) I think purists might quibble with the use of QTL to describe allelic imbalance mapping, and could also potentially cause confusion to readers not overly familiar with this field.

The term “QTL” is used in many different ways, which we agree can be confusing. We chose to use the definition of QTL given by the Complex Trait Consortium (Nat Rev Gen 2003): A genetic locus whose alleles affect variation in a quantitative trait. By this definition, our caQTLs are indeed QTLs for chromatin accessibility.

Reviewer #2:

In Tehranchi et al., 2018, the authors present a novel use of pooled sequencing applied to the identification of QTLs altering chromatin accesibility (caQTLs) by ATAC-seq. They demonstrate convincingly that their pooled approach can identify QTLs more efficiently than standard approaches, that CA is causally associated with transcription factor binding, long-distance chromatin interactions, and gene expression, and that chromatin accessibility is a likely molecular trait underlying GWAS variants. However, there are several concerns:1) caQTLs can only be detected if they affect their own read counts. Is this a major limitation of the method given the findings of previous studies? This should be addressed in the text.

We have added this to the Discussion: “There are several important limitations to our study. First, we have yet to map the majority of human caQTL variants—e.g. those specific to other cell types, those that affect CA of distal loci (in *trans* or long-range *cis*), or those involving rare variants or small effect sizes. In particular, our pooled approach can only map caQTLs that are covered by ATAC-seq reads; despite this limitation, our increased efficiency resulted in over 10- fold more caQTLs than previous studies (Degner et al., 2012, Kumasaka et al., 2016).”

2) Do caQTLs show different allele frequencies across genomic annotations (e.g. active TSS vs enhancers vs repressed regions, etc)? Does this contradict or support hypotheses of negative selection on caQTLs?

We appreciate this idea, and found that caQTLs near TSSs do indeed have lower minor allele frequencies than other caQTLs, as expected if they are under stronger purifying selection. We have added this to the Results: “…we also found that caQTLs near active TSSs have lower minor allele frequencies than elsewhere in the genome (median MAF = 0.17 for active TSS regions and 0.18 for flanking active TSS regions, compared to 0.21 for caQTLs elsewhere; Wilcoxon p = 1.5x10-28 and 8x10-6 respectively).”

3) Have the authors validated, in this or previous work, that they are able to reliably estimate the pre-ATAC/ChIP allele frequencies using their regression approach?

When originally developing the method (Tehranchi et al., Cell 2016), we performed simulations of a wide range of pool sizes and variant allele frquencies, and verified that the pre-ATAC/ChIP allele frequencies are accurately estimated by our method.

4) For the "Long-range interaction analysis," is potential mapping-bias of Hi-C reads taken into account by the authors?

We agree that for many applications of Hi-C data, mapping bias can be an important confounder. However, because we are using allele-specific read counts for this analysis, any bias that affects the mapping of both alleles will decrease the signal and make our results conservative. In order for a bias to cause a false positive signal, the more open alleles (with higher ATAC-seq read depth) would have to have a systematic bias towards having read pairs that falsely map at a greater distance than the closed alleles. We cannot imagine a plausible scenario where this could be the case. Nevertheless, we have used allele-specific inter-chromosomal read pairs from the same Hi-C data set as a control to test for any bias in our method, and did not observe any (Figure 4—figure supplement 1).

Minor Comments:1) Introduction section: while most GWAS variants are not in LD with coding variants or may be in LD with variants in regulatory regions, most genome-wide significant variants from GWAS aren't actually in regulatory regions.

We have changed this from “regulatory” to “noncoding”.

2) In "Characterizing fine-mapped eQTLs," the source data of the chromatin states (Ernst and Kellis, 2012) is present in the figure legend but is absent in the text and methods; adding a citation to one of these would be helpful

We have added this citation.

3) As a minor stylistic concern, chromHMM tracks have a standard color palette (e.g. red corresponds to promoter/TSS, orange/yellow to enhancer, green to transcribed regions, etc.) and this reviewer would prefer its use for Figure 3.

We chose an alternative color scheme for two reasons: we merged several annotations to simplify the figure, so there is no standard color for these merged annotations; and we chose colors to be discernable by readers who a red-green colorblind (for this reason we used green but not red in the figure).

4) It is not clear how read density was controlled for in subsection “Characterizing fine-mapped caQTLs”.

Thank you for pointing out this omission. We have added this paragraph to the Materials and methods section: “In Figure 3A we present the chromatin states of caQTLs compared to the genome as a whole. Much of the difference is driven by ATAC-seq read density; i.e. we would see strong enrichments for TSS and enhancers in any ATAC-seq experiment. To ask the more specific question of where caQTLs are enriched or depleted after controlling for read density, we selected non-caQTL variants matched for the number of reads to each shared caQTL, and then quantified the difference in chromatin state enrichments for shared caQTLs vs read-matched non-caQTLs. Details of read-matching are given in the “Effects of caQTLs on DNA shape” section below.”

5) The final paragraph of subsection “Characterizing fine-mapped caQTLs” references "trinucleotide" dependencies of CA variants but these are not mentioned in the text or Materials and methods section.

We have deleted this sentence (which referred to an analysis that was removed from the manuscript).

6) The figure legend for Figure 4 mentions using "allele-specific 3D chromosomal interaction" data; does this mean only heterozygous sites were considered? This should be described in more detail in the text and/or methods.

Yes, this is correct. We have added this to the Materials and methods section: “To obtain Figure 4A-B, we restricted the analysis to shared caQTLs (Fisher’s combined p < 5x10-6) with consistent directionality in a majority of the populations tested, and counted the number of Hi-C reads connecting each caQTL allele with any distal locus (>20 kb or >100 kb away). In order to use only allele-specific Hi-C reads, we restricted the analysis to read pairs where at least one read overlaps a heterozygous variant in the GM12878 cell line (as described in Rao et al., 2014). We then determined the number of caQTLs where the more accessible allele had significantly more Hi-C contacts (using a binomial test of allele-specific read counts), compared to the number where the less accessible allele was more interactive. See Supplementary file 1 for detailed results.”

7) Having the principal equations of the regression approach in the Materials and methods section would be helpful to readers.

We have included these.

8) The subsection "Effects of caQTLs on DNA shape" in the Supplementary Text includes analysis related to chromatin shape, which is not present in the main text.

We have added a summary of these results to the main text.

Additional data files and statistical comments:The provided data files, in addition to read data placed on public databases, should be sufficient.

[Editors' note: further revisions were requested prior to acceptance, as described below.]

The manuscript has been improved, but the reviewers remain concerned about the use of the term "fine-mapping" in the manuscript. Traditional fine-mapping, in the GWAS field for example, interrogates all (or the majority) of variants in a region, and assesses their relative evidence for causality based on association with a trait and relevant annotation. However, in your investigation, only a limited subset of variants in a region are considered, and the reviewers do not feel that "fine-mapping" is an appropriate description of the approach taken. The reviewers have suggested the following changes be made.

We respectfully disagree that our approach is not fine-mapping. In fact, most fine-mapping studies have been conducted using genotyping arrays targeted to loci of interest that do not capture the majority of variants in those loci. For example, the Cardio-Metabochip genotypes fewer than half of common variants (>5% frequency) in the targeted loci, and an even smaller fraction of rare variants [Morris et al., Nature Genetics 44:981–90 (2012)]. Since our approach is based on deep sequencing, the genotyping density in our study is actually higher within our regions of interest (specifically, the ATAC-seq peaks) than the genotyping density of fine-mapping microarrays in their regions of interest.

In addition, the definition of fine-mapping was recently given as “To refine the genomic localization of causal variants by the use of statistical, bioinformatic or functional methods.” [Schaid et al., Nature Reviews Genetics 19:491–504 (2018)] Our approach certainly fits this definition.

1) Remove the term "fine-mapping" from the title of the manuscript – this could be replaced by "determining" or "localising".

In light of the reasons described above, we have elected to retain the term “fine-mapping”.

2) Please provide further discussion of the limitation of the approach to localising causal variants. In particular, please include discussion to address the following comment from the initial review of your manuscript:"The authors describe their approach of prioritizing variants with significant allelic effects across populations as fine-mapping causal variants, which I think is not entirely accurate. The premise of fine-mapping is that most/all variants on in a region are evaluated for their effects on a trait; given that the authors are only able to map the subset of variants with sufficient ATAC-seq read depth there may be causal variants that, for example, map in a different site and affect chromatin accessibility across an entire region, map in the same site but on the edge and have low read depth, or map outside of a site entirely. While presumably the majority of variants with shared allelic effects are likely directly responsible for their imbalance, there are likely also variants where this isn't the case."

We have included this caveat in the Discussion: “Third, our fine-mapping approach is distinct from typical GWAS fine-mapping where many variants in a locus are genotyped in a large cohort; our pooled approach can only map caQTLs that are covered by ATAC-seq reads. Within these regions of open chromatin, our approach genotypes essentially all common variants, but variants outside these regions are not covered.”